# Spatio-temporal control of mitosis using light via a Plk1 inhibitor caged for activity and cellular permeability

Victoria von Glasenapp [1,2], Ana C. Almeida [3], Dalu Chang [2,4], Ivana Gasic [3], Nicolas Winssinger [2,4] ✉ & Monica Gotta [1,2] ✉

The ability to control the activity of kinases spatially and temporally is essential to elucidate the role of signalling pathways in development and physiology. Progress in this direction has been hampered by the lack of tools to manipulate kinase activity in a highly controlled manner in vivo. Here we report a strategy to modify BI2536, the well characterized inhibitor of the conserved and essential mitotic kinase Polo-like kinase 1 (Plk1). We introduce the same coumarin photolabile protecting group (PPG) at two positions of the inhibitor. At one position, the coumarin prevents the interaction with Plk1, at the second it masks an added carboxylic acid, important for cellular retention. Exposure to light results in removal of both PPGs, leading to the activation of the inhibitor and its trapping inside cells. We demonstrate the efficacy of the caged inhibitor in three-dimensional spheroid cultures: by uncaging it with a single light pulse, we can inhibit Plk1 and arrest cell division, a highly dynamic process, with spatio-temporal control. Our design can be applied to other small molecules, providing a solution to control their activity in living cells with unprecedented precision.

The development of chemical tools that can be spatially and temporally controlled is crucial to understand and characterize biological processes at the molecular level in living cells and organisms[1–4]. Light is a powerful tool for controlling molecular events and can be used to activate or deliver compounds[5–10]. The advantages of using light are its noninvasive nature, the ease of application and the exquisite spatial and temporal control. Photolabile protecting groups (PPGs) or photocages have been explored for several decades for application in research to control the action of small molecule inhibitors[11–13]. Such chemical tools are particularly useful to manipulate and study dynamic multi-step processes in cells. One example is mitosis, a critical phase of the cell cycle, in which a cell undergoes major morphological and biochemical reorganization to divide into two daughter cells that contain the same genetic information. Defects in mitosis can lead to diseases such as but not limited to cancer. Each step of mitosis must be therefore tightly and timely controlled to maintain the fidelity of cell division. This precise control is achieved through a complex system of regulatory proteins, amongst which kinases play a central role. New insights into how kinases regulate cell division continue to arise (e.g[14]). While some kinases control specific events during cell division[15,16], others are deployed throughout cell division. One such pleiotropic key regulator of cell division is the conserved Polo-like kinase 1 (Plk1)[17]. Plk1 controls several processes such as centrosome maturation, nuclear envelope break down, bipolar spindle assembly, chromosome segregation and cytokinesis through the phosphorylation of its many substrates (reviewed in refs. [18–21]). Moreover, Plk1 is an oncogene that is overexpressed in many cancers, such as lung cancer, breast cancer, and melanoma[22]. Inhibition of Plk1 results in mitotic arrest and aberrant cell division leading to cell death, making Plk1 an attractive target for cancer therapy[23].

[1]Department of Physiology and Metabolism, Faculty of Medicine, University of Geneva, Geneva, Switzerland. [2]NCCR Chemical Biology, University of Geneva, Geneva, Switzerland. [3]Department of Molecular and Cellular Biology, Faculty of Science, University of Geneva, Geneva, Switzerland. [4]Department of Organic Chemistry, Faculty of Science, University of Geneva, Geneva, Switzerland. ✉e-mail: nicolas.winssinger@unige.ch; monica.gotta@unige.ch

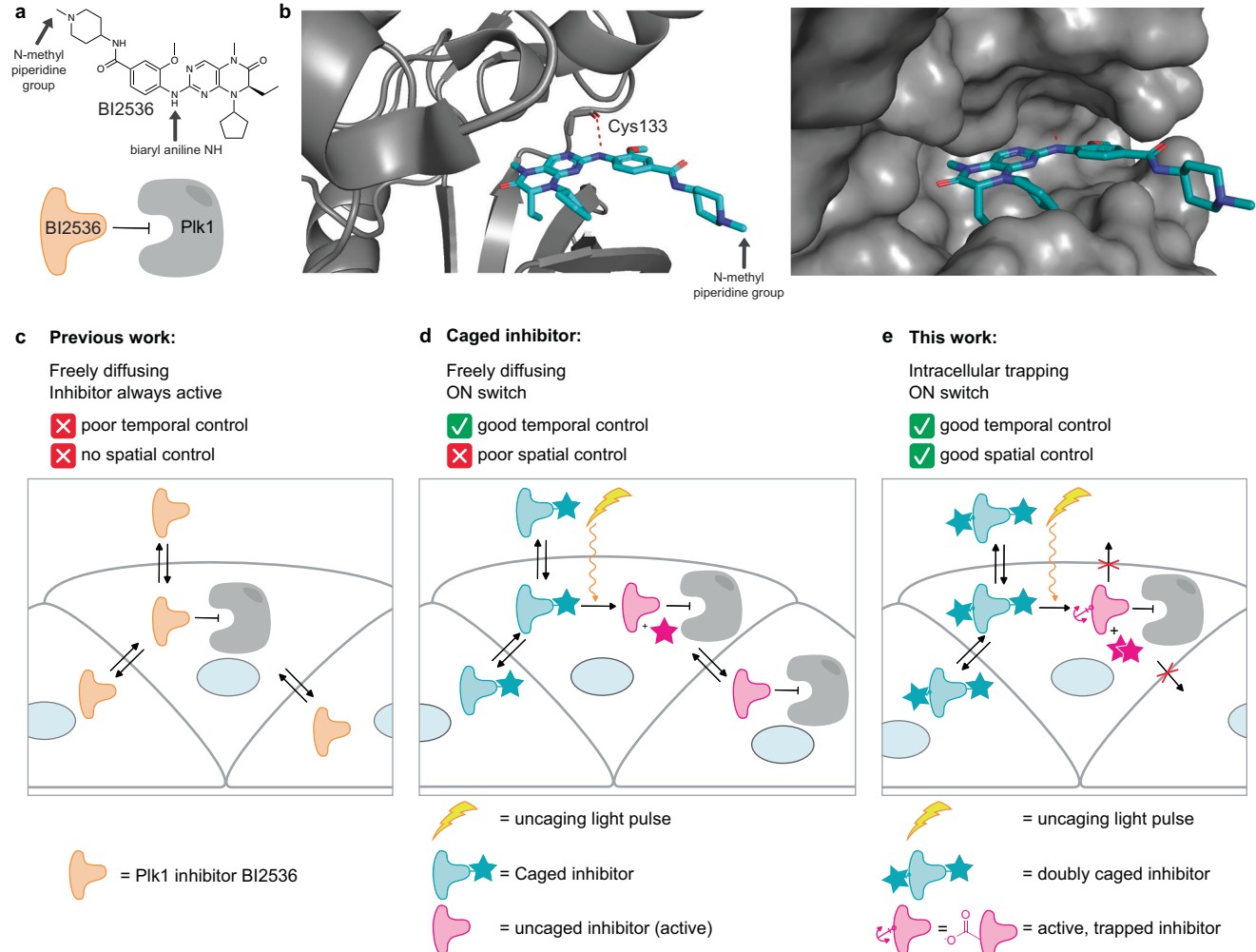

**Fig. 1 | Aim of this work. a** Top: Chemical structure of BI2536, arrows are pointing toward the methyl group on the piperidine ring and the aniline amine. Bottom: schematic representation of the interaction between BI2536 and Plk1. The same representation has been used in panels **c**–**e** and in other figures of article. **b** Co-crystal structure of Plk1 kinase domain (gray) with BI2536 (blue) (PDB 2RKU)[40]. Red dotted line: hydrogen bond between carbonyl of Cys133 of Plk1 with the aniline amine of BI2536. Left: protein is shown as a cartoon, right: protein is shown in the space-filling model. **c**–**e** Schematic representation of the spatial and temporal control of inhibition with the unmodified (**c**) and caged (**d**) and (**e**) Plk1 inhibitor. All schemes show a drawing of cells in culture. The cell is delimited by a grey line (cell membrane), blue circles are nuclei. In (**c**) the unmodified inhibitor (orange) is always active and diffuses freely between cells, resulting in poor temporal control (defined by the time of addition of the inhibitor) and no spatial control. In (**d**), the inhibitor caged for activity (pink with the cage represented as a pink star) can be activated with high temporal resolution using light (yellow lightning bolt and arrow) but has no spatial control as it diffuses, both in the caged and uncaged form, as shown by the arrow and the caged compound (blue) and the uncaged compound (pink) in the neighboring cells. This color code to label caged (blue), uncaged (pink) and unmodified inhibitor (orange) is used in all figures. In (**e**) the inhibitor is caged for activity and contains a moiety that renders the compound cell membrane impermeant when uncaged, which leads to temporally and spatially controlled inhibition.

The small molecule Plk1 inhibitor BI2536[24,25] (Fig. 1a, b) was initially developed for clinical applications but was unsuccessful in clinical trials mainly due to low response rate and hematological toxicity[26,27]. BI2536 specifically inhibits Plk1 with a 1000-fold selectivity over other kinases. It also inhibits other members of the Polo-like kinase family, Plk2 and Plk3, which are mostly active in interphase[24]. As a tool compound, BI2536 has been fundamental in the characterization of many functions of Plk1[28,29] as well as Plk2[30] and Plk3[31]. However, free diffusion of this inhibitor limits its applications in addressing research questions in a multicellular environment where spatial resolution of cell division inhibition is required (Fig. 1c).

One approach to enrich drugs into a subset of cells is to modify them with a moiety that will specifically bind to self-labeling protein tags such as the SNAP-tag[32] or the Halo-tag[33] expressed in the target population. Even though this method can lead to the accumulation of the compound in the cell expressing the protein tag, it requires the development of genetically modified cells or organisms, and it does not prevent the inhibitor from entering cells that do not express the tag, thus failing to achieve spatial resolution at the organismal level. Furthermore, it may confine the inhibitor in a suboptimal cellular environment (such as cytosol vs. nucleus).

An alternative method to increase intracellular retention of small molecules is to chemically modify them with an ester that is hydrolyzed by intracellular esterases to reveal a carboxylic acid[34–37]. However, this enzyme-catalyzed reaction cannot be externally controlled and depends on various factors, including the cell type to which the compound is applied and the molecular structure of the compound itself[38,39]. In addition, as in the previous example with small molecules binding to SNAP- and Halo-tags, the inhibitor, which is constantly active, can enter any cell in a multicellular setting and perform its function. This problem could be solved if the activity of the inhibitor was masked and could be unmasked with temporal control when the inhibitor has entered the target cells[36]. There is

therefore a need to generate new tools that enable the simultaneous spatial and temporal regulation of the activity and diffusion of small molecule inhibitors.

Here we have modified BI2536 to control its inhibitory activity spatially and temporally in living cells with a single light pulse. We have attached a coumarin PPG at two positions of the inhibitor: one photocage is attached at a position important for the binding to and inhibition of Plk1, enabling temporal control (Fig. 1d). The other coumarin photocage masks an added carboxylic acid at a different position, which, once unmasked, leads to cellular retention (Fig. 1e) and therefore restricts Plk1 inhibition to the cells having received the light pulse. Applying light to this caged BI2536 enables us to control progression through mitosis via Plk1 inhibition with exquisite temporal and spatial precision in cells grown in monolayer and spheroid cultures.

This approach can be extended to other small molecules where spatial and temporal regulation of their activity is required, opening new routes for controlled drug targeting in more complex systems.

## Results

### Attaching a coumarin photocage to BI2536 abrogates Plk1 inhibition

In this work we aimed at modifying BI2536 in a way to control both the activity and its cellular diffusion once activated (see Fig. 1). First, we investigated whether attaching a photolabile protecting group to BI2536 could be used to abrogate its activity and temporally control Plk1 inhibition (Fig. 1, a, b and d). Second, we introduced an anchoring moiety to spatially restrict the inhibitor by controlling cellular permeability (Fig. 1e and see below).

BI2536 binds to the kinase domain (Fig. 1b) as an ATP competitive inhibitor. It consists of a dihydropteridinone moiety, which, based on the co-crystal structure, forms hydrogen bonds with the kinase domain and an aminopiperidine at the solvent-exposed side of the nucleotide-binding pocket (PDB 2RKU, Fig. 1a and b, see arrows)[40]. The biaryl aniline NH of BI2536 is involved in a hydrogen bond network with the carbonyl of cysteine 133 of Plk1 (Fig. 1a and b), and is crucial for the binding of the inhibitor to the kinase[40,41]. We, therefore, chose this aniline to attach the photolabile protecting group to cage the activity.

As photolabile protecting group, we selected the 7-diethylaminocoumarin with the electron-donating p-ethoxystyryl moiety in the 3-position (Fig. 2)[42]. This caging group has an absorption that extends to 500 nm ($\lambda_{max}$ at 439 nm) with a high uncaging quantum yield $\Phi_u$ of 0.45 compared to 0.07 for the commonly used 7-diethylaminocoumarin (DEAC) cage ($\lambda_{max} < 400$ nm)[42]. This allows fast uncaging with short irradiation at a lower energy wavelength (including a 488 nm light) compared to DEAC[42,43], which is compatible with a light-sensitive process such as mitosis.

First, we synthesized BI2536 according to slight modifications from the reported protocol (the synthesis is shown in Supplementary Fig. 1)[44]. The crystal structure reveals that the methyl group on the piperidine ring (Fig. 1a, b) is not interacting with the kinase domain of Plk1, and previous studies have shown that this position can be modified without affecting the potency of the inhibitor[32,40,44,45]. We protected the piperidine amine with a tert-butyloxycarbonyl protecting group (Boc), which can be easily removed to further functionalize the inhibitor[44] (for example to induce intracellular retention as shown further below). This yielded the modified BI2536 Plk1 inhibitor (**Plk1i**, Supplementary Fig. 1). We attached the p-ethoxystyryl derivatized 7-diethylaminocoumarin cage to the aniline NH of the inhibitor via a carbamate (Fig. 2). To do this, we first treated **Plk1i** with triphosgene in basic conditions. In parallel, the alcohol on the coumarin was deprotonated with sodium hydride and added to the reaction mixture to yield the Plk1 inhibitor caged with the coumarin analogue **cPlk1i** with yields up to 63% (Fig. 2).

**cPlk1i** has an absorption maximum at 439 nm and an emission maximum at 535 nm (Fig. 3a), a property that we used to monitor the ability of the compound to penetrate cells. To quantify uncaging, we used irradiation of 455 nm light from a LED lamp source (8.3 mW/cm²) over a time-course of 2 minutes and monitored the products with ¹H NMR (Fig. 3b–d). NMR analysis revealed that full uncaging was achieved after 2 minutes (455 nm, 8.3 mW/cm²) (Fig. 3c, d and Supplementary Fig. 2a and b). Uncaging in water (Fig. 3e) and in PBS (Supplementary Fig. 3) was confirmed by LC-MS measurements with **cPlk1i** eluting at 2.54 minutes and the free inhibitor **Plk1i** eluting at 1.59 minutes. The apparent faster rate of uncaging in the LC-MS analysis compared to the NMR quantification is due to the larger area of irradiation in the LC-MS experiment. The light intensity that is successfully uncaged is comparable to intensities regularly used for GFP imaging, suggesting that these settings can be used to uncage the inhibitor under a microscope. Furthermore, **Plk1i** and **cPlk1i** were stable over 24 h in a cell culture medium in the presence of serum (Supplementary Fig. 4a–c).

We then assessed the effectiveness of the coumarin caging group to control inhibition. We would expect the caged inhibitor to have no effect on cell cycle progression, as the PPG should sterically hinder its activity. Once uncaged, the resulting Plk1 inhibition would lead to the accumulation of cells arrested in mitosis. We treated cells for 7 h with BI2536, **Plk1i** and **cPlk1i** and analyzed the cell cycle profile by flow cytometry (Fig. 4a, b and Supplementary Fig. 5). After treatment with the synthesized free inhibitor **Plk1i** (100 nM), 63.2% of cells were arrested in the G2/M phase of the cell cycle, which is comparable to the result obtained with the commercial inhibitor BI2536 (Fig. 4b) and the proportion of cells entering mitosis in the time course of the experiment (7 h)[25]. The results also corroborate that modification on the piperidine amine does not interfere with the activity of the drug, as

**Fig. 2 | Scheme of the synthesis of caged Plk1 inhibitor cPlk1i. a** triphosgene, DIPEA, THF, 0 °C, 30 minutes; **b** 1) **5**, NaH, THF, 0 °C, 30 minutes, 2) DIPEA, THF, 0 °C, 63% yield.

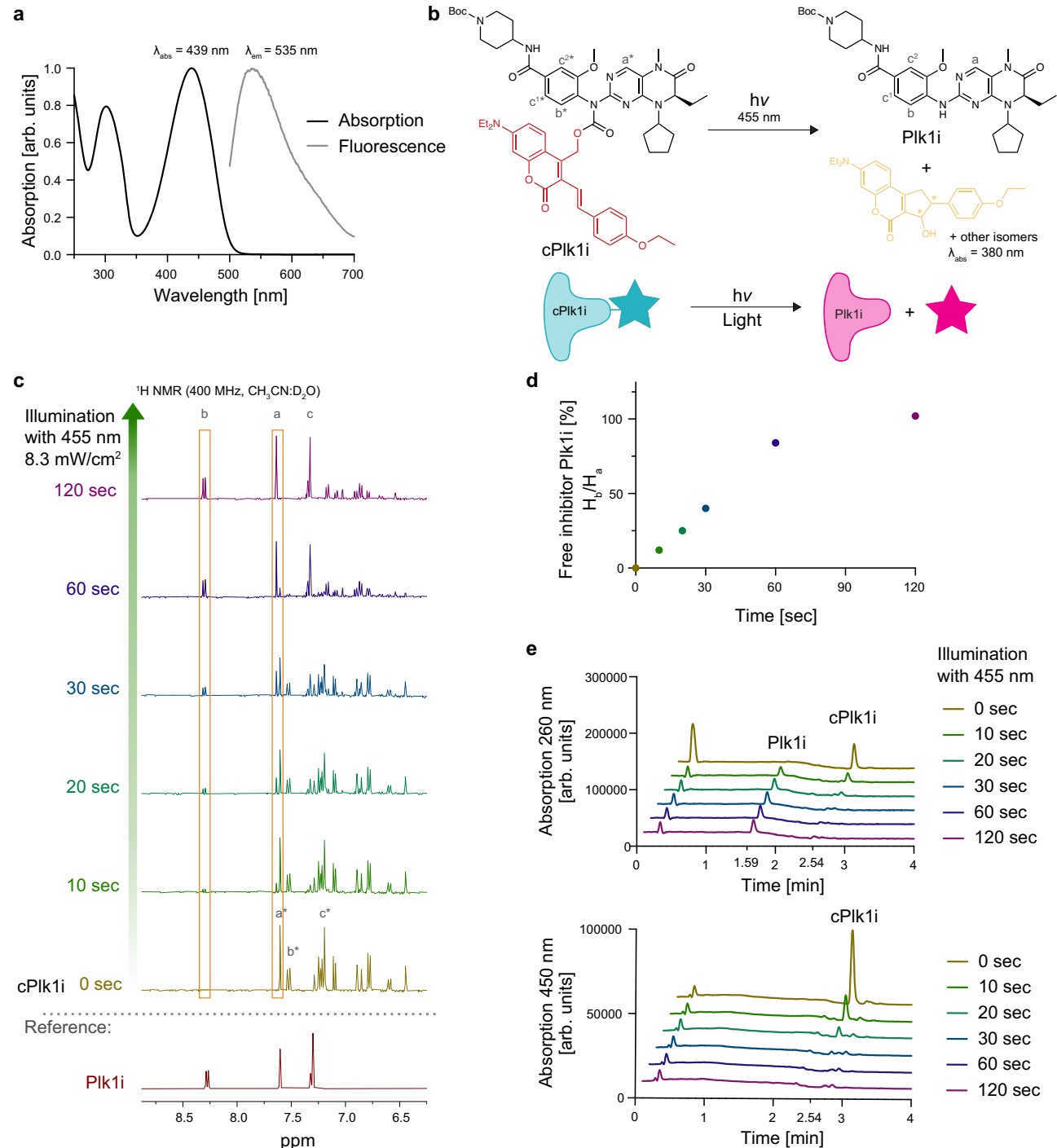

**Fig. 3 | Photochemical characterization of cPlk1i. a** Absorption and emission spectra of **cPlk1i. b** Uncaging reaction of **cPlk1i** releasing the active Plk1 inhibitor **Plk1i. c** Uncaging of **cPlk1i** monitored by ¹H NMR. The NMR sample (CH₃CN:D₂O) was illuminated with 455 nm light at 8.3 mW/cm² for the indicated times. The reference spectrum of **Plk1i** without the coumarin cage is below in red. Orange boxes: signals of protons a and b used to quantify uncaging, shown in (**d**). The assigned spectra are shown in Supplementary Fig. 2a and b. **e** Uncaging of **cPlk1i** monitored by LC-MS. Absorption chromatogram at 260 nm (top) and 450 nm (bottom). Source data are provided as a Source Data file.

shown previously[40,44] and the synthesized inhibitor bearing the Boc-protecting group efficiently inhibits cell cycle progression (Fig. 4a, b). **cPlk1i** remained inactive at concentrations up to 10 times higher than **Plk1i** as evidenced by the lack of accumulation of cells in G2/M compared to control treated cells (Fig. 4a, b). This shows that **cPlk1i** does not interfere with cell cycle progression (Fig. 4a, b).

The lack of a cell cycle arrest when cells are exposed to **cPlk1i** could be due to the inability of the caged inhibitor to enter the cells. To assess partitioning of the inhibitor inside cells, we took advantage of the fluorescent properties of coumarin (Fig. 3a). We imaged cells treated with **cPlk1i** (1 µM) and observed fluorescence in the cytoplasm (Fig. 4c, d) indicating that the inhibitor partitioned favorably inside cells and the lack of cell cycle arrest was not due to failure of the drug to enter the cells. Furthermore, we investigated whether the compound is colocalizing with lysosomes or endosomes. We could not observe colocalization with those

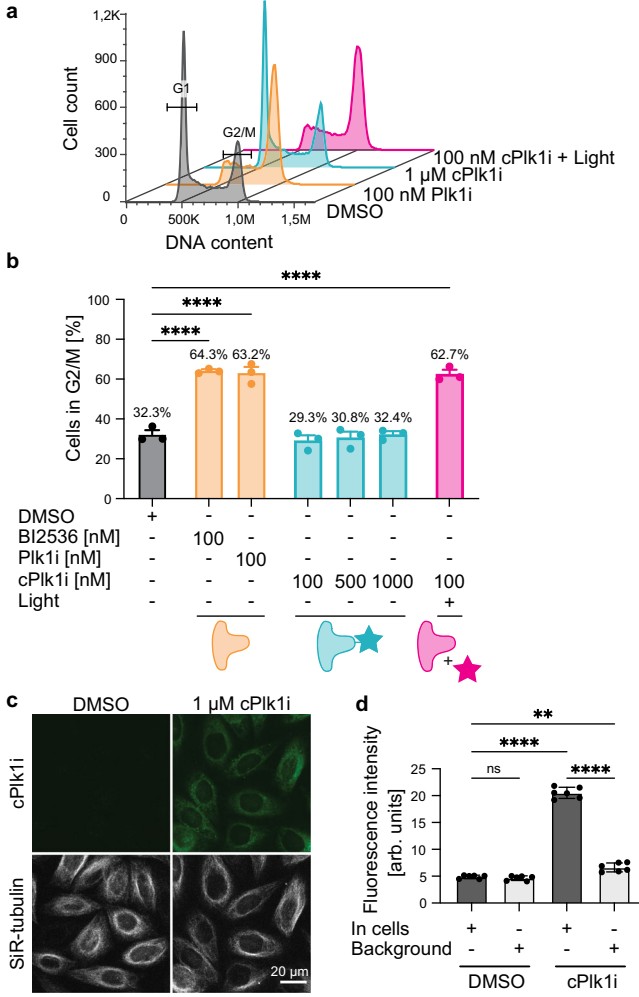

**Fig. 4 | cPlk1i does not interfere with cell cycle progression. a** Cell cycle profiles of 4 of the conditions shown in **b** of HeLa cells after 7 h of treatment with the free inhibitor **Plk1i** and the caged inhibitor **cPlk1i** (- and + uncaging). The cell cycle profile of the remaining conditions plotted in **b** are shown in Supplementary Fig. 5. **b** Plot of G2/M content of the cell cycle profiles after treatment of HeLa cells for 7 h with the indicated inhibitors. Graph shows mean ± standard error of the mean (SEM). Ordinary one-way ANOVA was performed comparing each condition to the DMSO control. $N = 3$, with 20'000 cells per condition per experiment. Each dot represents the percentage of one experiment. **c** and **d** Microscopy picture showing fluorescence from **cPlk1i** (1 μM) (top) and SiR-tubulin (25 nM) for reference (bottom) (**c**) and quantifications (**d**). Shown is the mean intensity ± standard deviation (SD), each dot represents average intensity of all cells analyzed per image, $n = 272$, $N = 1$, 6 technical replicates. Ordinary one-way ANOVA was performed. For all panels no label or ns means not significant: $p > 0.8$ (exact $p$ values are provided in the Source Data file), **: $p = 0.002$, ****: $p < 0.0001$. Source data are provided as a Source Data file.

organelles suggesting that the compound enters by passive diffusion (Supplementary Fig. 6a, c). Next, we tested if **cPlk1i** is active after in vitro uncaging: 62.7% of the cells treated with 100 nM of **cPlk1i** previously uncaged (455 nm, 2 min, 8.3 mW/cm²) arrested in G2/M, in agreement with the results obtained using BI2536 and **Plk1i** (Fig. 4a, b). We did not observe an increase in cell death in cells treated with **cPlk1i** neither before nor after uncaging (Supplementary Fig. 7a, b), indicating that the compound is not toxic. In conclusion, our data show that the newly synthesized inhibitor is inactive when caged and that uncaging restores its activity and leads to arrested cell division.

## Uncaging cPlk1i *in cellulo* results in cell division arrest

We next tested whether the caged inhibitor could be uncaged *in cellulo* with the optical setup of a wide field microscope. The intensity of the 488 nm light under the microscope was 5.5 mW/cm², which is comparable to the LED irradiation used previously and should afford uncaging of **cPlk1i** within minutes. As treatment with 1 μM **cPlk1i** did not result in toxicity (Supplementary Fig. 7b), we decided to increase the concentration of **cPlk1i** compared to BI2536. This allowed us to use a shorter uncaging light pulse, since only a portion of the inhibitor needs to be uncaged to reach the biologically relevant concentration.

Since uncaging leads to an intramolecular cyclization[42] (Fig. 3b), which changes the π-system and blue-shifts the absorption of the coumarin, the quantum yield of fluorescence at 535 nm decreases and the fluorescence decrease can be used to monitor uncaging. We treated cells with **cPlk1i** (500 nM), illuminated them for 50 seconds and used the fluorescence decrease of the coumarin after illumination as a proxy of uncaging. Illumination resulted in about 43% reduction in fluorescence intensity, indicating that **cPlk1i** can be uncaged under the microscope. Under these conditions (5.5 mW/cm², 488 nm irradiation), the half-life of photolysis is calculated at 64 seconds (Supplementary Fig. 8).

To test the efficiency of Plk1 inhibition, we first synchronized cells with monastrol, an inhibitor of the motor protein Eg5, which is essential for centrosome separation and bipolar spindle assembly during early mitosis[46]. Inhibition of Eg5 arrests cells in mitosis with monopolar spindles. Removal of the Eg5 inhibitor will allow cells to complete mitosis[46], which is prevented if Plk1 is inhibited. We released cells arrested with monastrol in DMSO, in **cPlk1i** containing medium with or without light and, as a positive control of the inhibition, in BI2536 or **Plk1i** (Fig. 5a). To facilitate the analysis of arrested cells, we stained them for the kinetochore protein BubR1, which remains on kinetochores when Plk1 is inhibited[47]. When we released cells in DMSO, with or without light exposure, cells progressed through mitosis (Fig. 5b, c), indicating that illumination per se does not arrest cell division. When we released cells in the presence of BI2536 (100 nM) and **Plk1i** (100 nM), cells failed to proceed through mitosis and remained arrested with BubR1 positive kinetochores, indicating that Plk1 is inhibited (Fig. 5b, c). In cells released from Eg5 inhibition in the presence of **cPlk1i** (500 nM) and kept in the dark, no increase of arrested cells was observed (Fig. 5b, c). When cells were released in the presence of **cPlk1i** and illuminated with 488 nm light, they remained arrested in mitosis as shown by the positive BubR1 staining (Fig. 5b, c). The inhibition and efficiency of **cPlk1i** treatment were independent of the cell synchronization, as shown by the increased number of mitotic cells found over 6 h when cells are treated with **cPlk1i** and illuminated (Supplementary Fig. 9).

These data indicate that **cPlk1i** can successfully be uncaged with temporal resolution under a microscope and arrest cell division.

When inhibition in specific cells is required, one problem of small molecules is their ability to diffuse fast. We asked whether the activation of **cPlk1i** was spatially restricted. As above, cells released from a monastrol arrest were treated with **cPlk1i** (1 μM). We uncaged **cPlk1i** in a field of view in the well with a single light pulse of 2, 6 or 12 seconds and imaged both the illuminated field of view and a field 6 mm away (Fig. 5d and e show cells exposed to 12 seconds light, 2 and 6 seconds exposure are shown in Supplementary Fig. 10). Cells were arrested independently of the distance to the uncaging region, suggesting that the uncaged inhibitor diffuses in the well. In conclusion, although **cPlk1i** can be successfully activated *in cellulo* under a microscope resulting in inhibition of cell cycle progression with temporal resolution, it can diffuse and inhibit neighboring cells.

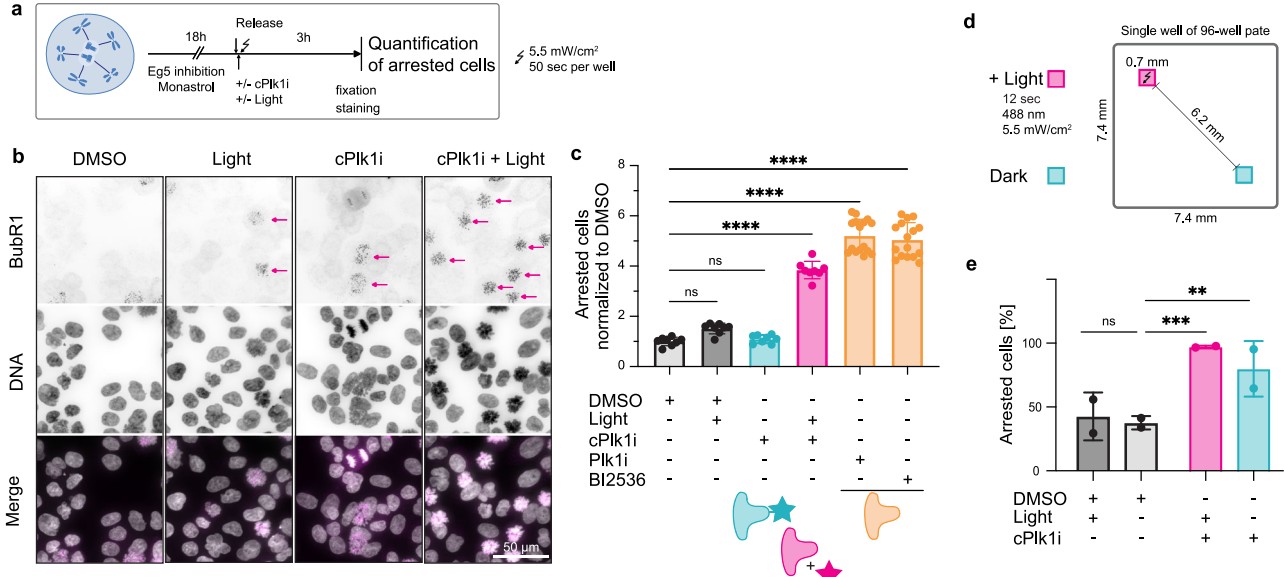

**Fig. 5 | cPlk1i can be uncaged under the microscope. a** Overview of the experimental setup. **b** Representative microscopy images of cells treated with control or **cPlk1i**, − or + light, quantified in **c**. Arrows point towards BubR1 positive nuclei. DNA was visualized with Hoechst. Scale bar 50 μm. **c** Quantification of arrested cells after release (see materials and methods for the quantifications). Cells were treated with **cPlk1i** (500 nM), **Plk1i** (100 nM), BI2536 or DMSO (100 nM). Dots represent the technical replicates of the two experiments, number of cells $n = 641'873$, $N = 2$. Shown is the mean ± SD. **d** Schematics of exposure to light and quantification of two fields of view (FOV) in a single well containing HeLa H2B-mCherry cells[66]

arrested with monastrol and released in **cPlk1i** (1 μM) or DMSO. **e** Quantification of arrested cells 120 minutes after release from monastrol arrest. The number of arrested cells at 120 minutes is normalized to the number of arrested cells at time 0 after monastrol release. DMSO control is the same as in Fig. 6b and Supplementary Fig. 10 as both inhibitors were tested in the same experiments. Shown is the mean ± SD. Number of cells $n = 4059$, $N = 2$. Ordinary one-way ANOVA was performed in panels (**c**) and (**e**). ****: $p < 0.0001$, ***: $p = 0.001$, **: $p = 0.0094$, ns: $p \geq 0.18$ (exact p values are provided in the Source Data file). Source data are provided as a Source Data file.

## Uncaging a carboxylic acid results in cellular retention

To further improve spatial resolution, we aimed at trapping the inhibitor inside the cell once activated. One approach is to introduce esters that will be enzymatically hydrolyzed inside the cell to give rise to a carboxylic acid which will be negatively charged and therefore restrict membrane permeability[35,36]. We selected the methylester and attached it to the inhibitor at the piperidine amine, which is at the solvent exposed side of the inhibitor (Fig. 1a and crystal structure Fig. 1b). For streamlined synthesis we first introduced an alkyne at the piperidine amine to then click the methyl-2-azidoacetate onto to the **Plk1i-Alkyne** inhibitor by copper-catalyzed cycloaddition[48,49] (synthesis in Supplementary Fig. 11), giving rise to **Plk1i-methylester**. We asked if the ester modification interferes with the activity of the inhibitor. When applied without washing, **Plk1i-methylester** (100 nM) arrested cells, indicating that the modification does not interfere with the activity of the inhibitor (Supplementary Fig. 12a, b). To test the retention inside cells, we treated with the inhibitor for 3 h to allow for the hydrolysis reaction to take place[34]. We then washed to remove the excess inhibitor and incubated overnight (Supplementary Fig. 12c). Washing resulted in the loss of Plk1 inhibition, as determined by the number of arrested cells in the dish (Supplementary Fig. 12c). These results suggest that the methylester modified Plk1 inhibitor is not efficiently hydrolyzed in the cells, and it is not retained in the cytosol.

Instead of relying on an enzymatic reaction for the unmasking of the carboxylic acid, which we cannot control, we decided to introduce a second coumarin photocage that, once removed with light, will trap the inhibitor inside cells (Fig. 1e and Fig. 6a). We therefore caged the carboxylic acid through an ester linkage that will be cleaved upon light irradiation releasing the carboxylic acid modified Plk1 inhibitor. As PPG we selected the same coumarin that we used to cage the activity of the Plk1 inhibitor which allows, with one light pulse, simultaneous uncaging of the activity of the inhibitor and of the carboxylic acid promoting retention of the active inhibitor in the cell.

To synthesize the doubly caged inhibitor we started with the **Plk1i-Alkyne** and attached the coumarin cage at the aniline amine as previously shown (Supplementary Fig. 13). Briefly, we activated the aniline amine with a chloroformate and reacted it with the alkoxide of the coumarin (Supplementary Fig. 13). We further caged azidoacetic acid with the *p*-ethoxystyryl derivatized 7-diethylaminocoumarin cage by esterification using EDC activation. In the last step we clicked the caged carboxylic acid to the caged inhibitor **cPlk1i-Alkyne** by copper catalyzed azide-alkyne cycloaddition[48,49] to obtain **cPlk1i-COOc** (Supplementary Fig. 13 and Fig. 6a). In vitro the doubly caged inhibitor **cPlk1i-COOc** was fully uncaged with 30 seconds illumination (455 nm, 15 mW/cm²) in water (Supplementary Fig. 14a) and in PBS (Supplementary Fig. 14b) and it was stable over a course of 24 h in cell culture medium supplemented with serum (Supplementary Fig. 4a, d). Similar to **cPlk1i**, **cPlk1i-COOc** accumulated inside cells but did not colocalize with endosomes or lysosomes (Supplementary Fig. 6b, c).

To assess whether the functionalization with the carboxylic acid retained the Plk1 inhibitory function of our compound, we performed four assays. First, **cPlk1i-COOc** treatment and activation resulted in an increase in mitotic cells over the course of 6 h similar to treatment with BI2536 (Supplementary Fig. 9). Second, cells treated with **cPlk1i-COOc** and activated arrested with monopolar spindles (Supplementary Fig. 15a) and centrosome maturation was impaired, quantified by γ-tubulin intensity at the spindle pole (Supplementary Fig. 15b), as shown for BI2536 (e.g.[24]). Third, we tested whether treatment with **cPlk1i-COOc** displaces Plk1 from kinetochores, as shown for BI2536[24] (Supplementary Fig. 15d, e). Activating **cPlk1i-COOc** in cells resulted in the displacement of Plk1 from kinetochores (Supplementary Fig. 15d, e). Fourth, we investigated whether the phosphorylation of BubR1, which is a known direct target of Plk1 and becomes phosphorylated by Plk1 on Serine 676[50], was decreased following treatment with our compound. Inhibition with BI2536 as well as treatment with **cPlk1i-COOc** and activation with light led to a decrease in S676 phosphorylation

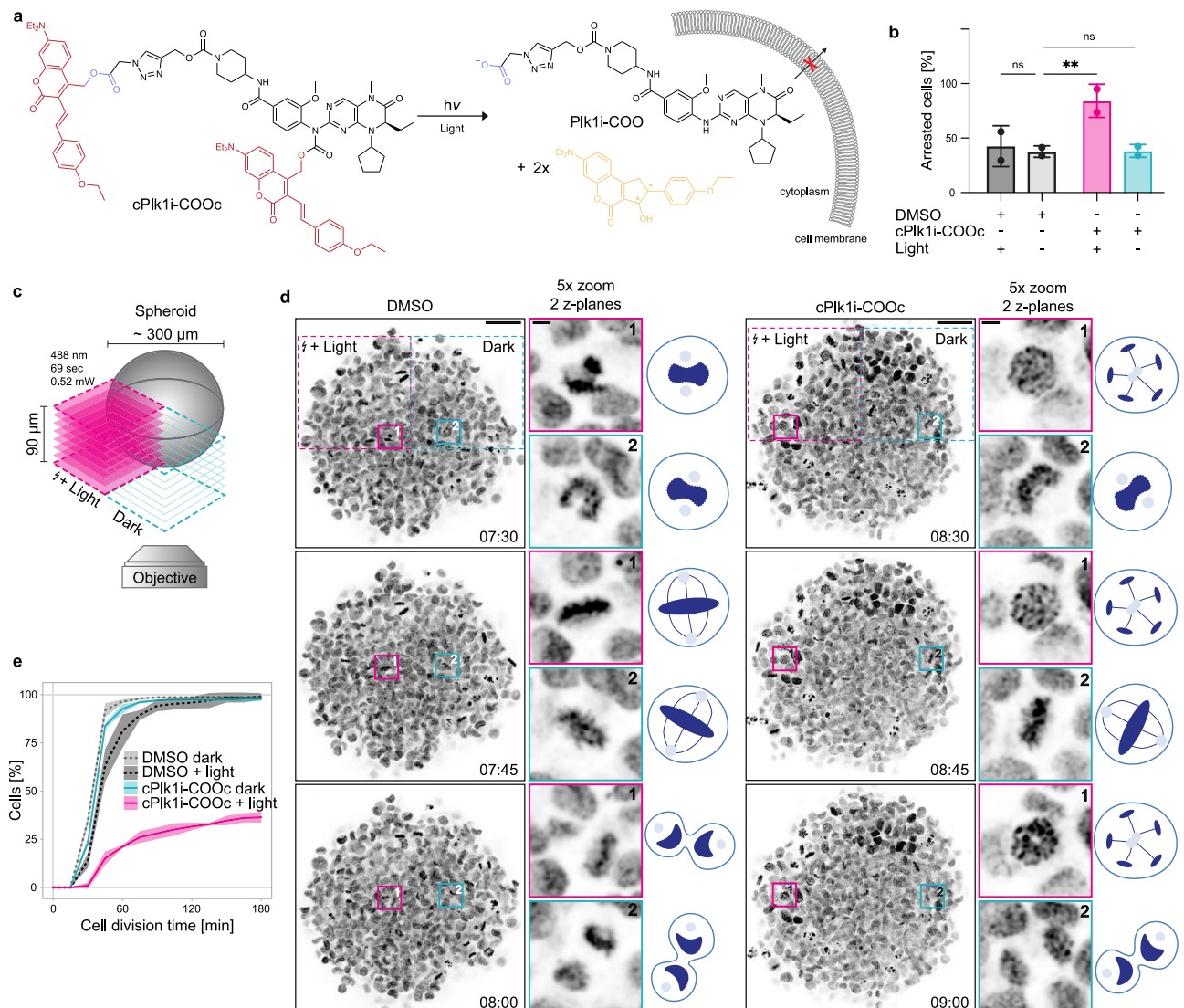

**Fig. 6 | cPlk1i-COOc allows spatial control of Plk1 activity in two and three-dimensional cell culture. a** Structure and uncaging reaction of **cPlk1i-COOc**. **b** Quantification of arrested cells in the experimental set up and conditions shown in Fig. 5d, e. Cells were treated with DMSO or **cPlk1i-COOc** (1 μM). DMSO control is the same as in Fig. 5e and Supplementary Fig. 10 as both inhibitors were tested in the same experiments. Graph shows mean ± SD. Number of cells $n = 4037$, $N = 2$. Each condition was compared to the dark DMSO control of the respective condition by performing ordinary one-way ANOVA. **\*\***: $p = 0.0033$, ns: $p \geq 0.999$ (exact p values are provided in the Source Data file). **c** Scheme of the experimental setup in spheroids. **d** Confocal images of consecutive time frames of stably expressing H2B-mCherry (black) spheroids treated with DMSO or **cPlk1i-COOc** (5 μM). The pink dotted line box on the left highlights the illuminated quadrant of the spheroid, as shown in the scheme in panel **c**, the cyan dotted line box on the right marks the dark quadrant. The smaller solid-line boxes within the illuminated and dark quadrants represent examples of individual cells. Magnified views of the solid-line boxes are shown on the right of the spheroid. Time points indicated in hh:min. Scale bar 50 μm, scale bar in the 5x zoom 5 μm. **e** Cumulative frequency plot of mitotic duration. The dotted or full lines show the mean, the shaded areas are the SD, $n = 983$, $N = 3$. The Mann-Whitney test was used to compare medians (Supplementary Table 1). Source data are provided as a Source Data file.

(Supplementary Fig. 15f, g). These results indicate that the functionalization does not abrogate the Plk1 inhibitory function of our compound.

To test whether **cPlk1i-COOc** allows to control Plk1 activity in a specific field of view, we implemented the same assay as for **cPlk1i** (Fig. 5d, e). We treated cells released from Eg5 inhibition with **cPlk1i-COOc** (1 and 5 μM) and uncaged in a defined field of view for up to 12 seconds (5.5 mW/cm², 488 nm) (Fig. 6b and Supplementary Fig. 10). While the 2 and the 6 seconds light pulse were not sufficient to arrest cells in mitosis with a concentration of 1 μM **cPlk1i-COOc** (Supplementary Fig. 10), the 12 seconds light pulse led to cells that arrested in mitosis (Fig. 6b). Increasing the concentration of **cPlk1i-COOc** to 5 μM led to arrested cells with the 2 seconds of irradiation (Supplementary Fig. 10). Strikingly, cells in the same well that did not receive the

uncaging light pulse were able to divide and did not show signs of mitotic arrest (Fig. 6b, Supplementary Fig. 10), corroborating that **cPlk1i-COOc** was uncaged, inhibited Plk1 and was retained in the cells where it was uncaged. The increased concentration of the compound did not result in cell death neither before nor after uncaging (Supplementary Fig. 16).

To further explore the potential of controlling Plk1 inhibition in specific cells, we grew human cells in three-dimensional cellular aggregates - spheroids[51,52]. We treated the spheroids for 2 h with DMSO or the **cPlk1i-COOc** (5 μM) (see Supplementary Fig. 17a, b for compound penetration in the spheroids). Subsequently, we illuminated one quarter of the spheroids by confocal laser scanning microscopy with 488 nm light for approximately one minute, and performed live imaging to follow cell division, taking an image stack every 15 minutes

for 12 h. Mitotic duration was determined by monitoring the time from mitotic entry (increased chromosome condensation) to anaphase onset (chromosome separation) in illuminated versus dark quadrants (Fig. 6c–e, Supplementary movies 1–3). In the DMSO-treated spheroids (Fig. 6d, e, Supplementary movie 1), cells completed mitosis within 45 minutes, regardless of light exposure, indicating that the uncaging conditions did not compromise cell cycle progression. In the spheroids incubated with the **cPlk1i-COOc** inhibitor (Supplementary movies 2 and 3), cells in the quadrant not exposed to light divided within the same time as the DMSO treated cells, whereas in the quadrant exposed to light, 70% of cells that entered mitosis were unable to exit. Arrested cells were present in both the outer layer and the core of the spheroid (Supplementary Fig. 17c), suggesting that the compound penetrates and is activated to sufficient levels to inhibit cell division even in the spheroid's core. These data demonstrate that the doubly caged Plk1 inhibitor, **cPlk1i-COOc**, can effectively block cell division in a 3D setting within a pre-defined region, without globally disrupting normal cell cycle progression.

## Discussion

Here we developed a light-controlled modified BI2536 Plk1 inhibitor to arrest cell division with exquisite spatial and temporal control. This was achieved by adding the same photolabile protecting group to two different positions on the small molecule. Addition of the photocage on one position, the biaryl aniline amine, abolished the interaction of BI2536 with Plk1, rendering the compound inactive until uncaged. The other photocage masked a carboxylic acid that we introduced on the inhibitor, trapping the inhibitor inside the cell upon uncaging. As photocage, we used a coumarin analogue with an extended aromatic system which enabled rapid uncaging with short light pulses at wavelengths of up to 488 nm[42], compatible with light sensitive processes such as mitosis.

By applying this compound to spheroids, we selectively inhibited cell division in the area of the spheroid that was exposed to the uncaging light, while the neighboring cells continued dividing normally.

The design we developed offers several main advantages. First, the modified inhibitor is inactive until uncaged with a light pulse. Such a feature allows for the activation of the inhibitor with high temporal control, which is particularly useful for teasing apart the different roles of kinases in multi-step processes such as mitosis. Second, using the same coumarin photocage allows unmasking of the added carboxylic acid in one step, bypassing the need for a second light pulse. The uncaged carboxylic acid traps the inhibitor inside the cell, preventing its diffusion to neighboring cells and allowing spatial control in a multicellular setting. Masking a carboxylic acid has been done previously by introducing different esters, which can be hydrolyzed by an enzyme-catalyzed reaction[35,36]. The limitations of this method, which was not successful in the present case, are that the hydrolysis cannot be controlled and relies on intracellular levels and specific groups of esterases that are differentially expressed across cell types[34,38]. Third, this caged inhibitor can be applied to any cell type with no need of specific genetic modifications or transgenic expression of self-labeling proteins. Fourth, the PPG that we used can be cleaved using light in the visible spectrum, which does not interfere with cell viability nor light sensitive processes such as mitosis at the dose required for uncaging. One limitation of photolabile protecting groups is the penetration in tissues of the light required to uncage. While light penetration was not a limiting factor in the present study, it has been shown that the coumarin that we used can be cleaved with two-photon irradiation with light of 730–880 nm[42], which can overcome light penetration limitations. However, two-photon activation also presents challenges such as small irradiation area and significant heat generation. Alternatively, PPGs activatable with longer wavelengths would be beneficial for further biological applications. This highlights the need to continue to develop photolabile protecting groups that are compatible with biological processes in many organisms[43,53,54].

The design that we use here should be broadly applicable to small molecule drugs provided structural information is available to identify a solvent exposed area of the molecule that can be modified to include a caged acid. Kinase inhibitors lend themselves particularly well because this class of inhibitors often contains a morpholino group or other saturated heterocycles that are solvent exposed and tolerant to modification, in addition to an aniline that participates in important interaction with the nucleotide binding site and can be caged[55,56]. More generally, with the improvement of in silico tools to model the interaction of compounds with their targets, many other small molecules could be modified in a similar manner and tested for their activity and cellular retention.

Small molecules with this kind of modifications allowing spatiotemporal regulation will be useful to control and study cellular processes in 3D settings such as organoids or mammalian embryos allowing to understand how their targets contribute to tissue formation and embryonic development[57,58]. From a therapeutic standpoint, there is growing interest in photopharmacology[5,59–65]. The present work adds another element of control, the retention of the drug at the site of uncaging.

Our study paves the way to the design of modified small molecules whose activity can be precisely controlled using just a few second of visible light irradiation with prolonged duration by virtue of cellular retention.

## Methods

### Absorption and fluorescence emission spectra

Absorption spectra were measured using a Jasco v-650 spectrophotometer, equipped with a deuterium and halogen lamp. Fluorescence emission was measured using Molecular Devices Spectra Max M5 spectrometer or a TECAN SPARK multimode microplate reader in PBS.

### In vitro photouncaging

To monitor the uncaging of the inhibitor over time by NMR, the caged inhibitor was dissolved in $ACN/D_2O$ 1:1 and irradiated for the indicated times. The sample was illuminated with a collimated LED light 15 cm above the sample (455 nm, 1 W: Thorlabs, part numbers M455L2 S/N M0027835, COP-1A) ($8.3\,mW/cm^2$ or $15.5\,mW/cm^2$). The intensity of the LED lamp with the different collimators was measured with a PM100USB power meter and S-170c power sensor (Thorlabs). An [1]H NMR spectrum was taken at the indicated time points. Assigned NMR spectra of **Plk1i** and **cPlk1i** are shown in Supplementary Fig. 2a and b. For monitoring by LC-MS, a 50-100 µM solution in $H_2O$ (10% DMSO) or PBS (10% DMSO) of the caged compound was prepared and irradiated with the same lamp for the indicated time. After every time point an LC-MS spectrum was taken. For the quantification, the signal of proton **b** of the inhibitor shifts from 7.1 ppm to 8.3 ppm after uncaging. The integral increases with longer irradiation times of 455 nm light, showing the increased amount of free inhibitor. This signal was normalized to the proton **a** at 7.6 ppm, which shifts by 0.02 ppm during uncaging. To activate the caged inhibitor for application in the cellular system, a 100 µM solution in $H_2O$ (10% DMSO) was prepared and activated using the same lamp for 90 seconds.

### Cell culture and drug treatment

Cells were cultured in Dulbecco's modified medium (DMEM; Gibco, 41965-039) supplemented with 10% FCS (Fetal Calf Serum, Biowest, S0750-500), 2 mL L-glutamine (Sigma, G7513), 100 units/mL penicillin and 100 µg/mL streptomycin (Pen/Strep) (Thermofisher). Commercial drugs and synthesized compounds were stored as a 5 mM or 10 mM stock solution in DMSO (ITW reagents) at −20°C. For drug treatments drugs were diluted into the indicated cell culture medium with a

maximum concentration of 0.5% DMSO. The commercial drugs were used at the following concentrations: BI2536 100 nM (opnMe, Boehringer Ingelheim and Selleckchem), monastrol 100 µM (GLPBio, GC14929). Synchronization with Eg5 inhibition was performed for 18 h with monastrol. To prevent premature activation of the caged inhibitors, the plates were wrapped in aluminum foil and kept in the dark throughout the incubation period. HeLa K cells (atcc, CCL-2), HeLa H2B-mCherry/MTS-GFP cells (Kind gift of Izabela Sumara)[66] (abbr: HeLa H2B-mCherry) were used in this study.

In Figs. 5, 6b, Supplementary Figs. 7, 9, 10 and 16 16.000-20.000 (150 µL/well) cells were seeded in µ-plate 96-well black (ibidi, 89626) in Fluorobrite (Gibco, A18967-01) supplemented with 10% FCS, 2 mM L-Glutamine 42 h before experiment. In the experiments in Figs. 5, 6b and Supplementary Fig. 10, the next day, 18 h before the experiment, the cells were treated with monastrol (100 µM) in Fluorobrite to arrest cells in metaphase with monopolar spindles. Cells were then released from the mitotic block using a peristaltic based multimode dispenser (Multiflow FX), to avoid washing off the mitotic cells. The medium was gently exchanged five times leaving 135 µL at the end of the wash with the peristaltic based multimode dispenser. After the release, cells were treated with the indicated drugs by adding 15 µL of a 10x solution in Fluorobrite to 135 µL of medium in each well.

## Cell cycle analysis by flow cytometry

To analyze the cell cycle by flow cytometry, 500'000 HeLa K cells were seeded in DMEM with 10% FCS and Pen/Strep and incubated overnight at 37°C, 5% CO$_2$. The next day the medium was replaced with medium containing the drugs at the given concentrations. After incubation for 7 h, cells were collected, washed with PBS (phosphate buffered saline) and fixed with ethanol (70%) overnight at −20°C. After fixation, cells were washed with PBS, then stained with Propidium Iodide (PI/Rnase, 1022195, BD Biosciences) for 15 minutes. Stained cells were analyzed using an Accuri C6 flow cytometer (BD Bioscience) equipped with 2 lasers and operated using BD Accuri C6 software. Cell cycle profiles were analyzed using FlowJo™ software and the statistics were performed using GraphPad Prism. The gating strategy is exemplified in Supplementary Fig. 5.

## Uptake of cPlk1i

HeLa K cells were seeded in µ-Slide 8 well dish (ibidi, 80826) in DMEM with 10% FCS and Pen/Strep and incubated overnight at 37°C, 5% CO$_2$. The next day cells were treated with SiR-tubulin (25 nM, Spirochrome) to visualize the microtubules in imaging medium (Leibovitz L-15, Gibco 21083-027, with 10% FCS) and after 4 h, with **cPlk1i** (1 µM). Images were acquired immediately after **cPlk1i** (1 µM) treatment with an A1r microscope (Nikon) equipped with 405 nm 100 mW laser, 488 nm 50 mW laser, excitation filter 472/30 nm and emission filter 520/35 nm and a 640 nm 40 mW laser, excitation filter 628/40 nm and emission filter 692/40 nm and a DU4 detector, using a Plan Apo VC 60x (NA 1.4) objective. The average intensity projection of the images was quantified. A mask of the cell was obtained based on the tubulin signal and the fluorescence intensity of the coumarin was quantified for each cell within this mask. The mean intensity per cell was normalized to the DMSO treated control. Images were processed and analyzed using the Fiji (ImageJ)[67].

## Uncaging under the microscope – 2D cell culture

In Fig. 5b, c, Supplementary Figs. 7–10, an IXM confocal automatic microscope (Molecular Device) with 10X objective (NA 0.45, Plan Apo Lambda) was used for uncaging. Activation of the inhibitor was done with a 488 nm light source focused through the objective. The light has an intensity of 5.5 mW/cm$^2$ at the sample. Activation was performed for 50 seconds with 100% light power scanning 25 fields of view per well for 2 seconds each.

To quantify the fluorescence coming from the coumarin cage in Supplementary Fig. 8, one picture was taken with 10 ms illumination time before uncaging and after uncaging and the total fluorescence for each picture was quantified.

## Immunofluorescence and imaging

In Fig. 5b and c, after treatment and activation cells were incubated for 3 h before fixation. For fixation, the medium was removed using the multichannel pipette and tilting the plate to nearly 90 degrees. Then 100 µL fixation medium (20 mM PIPES (pH = 6.8), 10 mM EGTA, 1 mM MgCl$_2$, 0.2% Triton X-100, 4% formaldehyde) were added and incubated for 10 minutes at 37°C. After fixation the sample was washed twice with PBS and incubated for 1.5 h with anti-BubR1 mouse primary antibody (1:20, ABCD antibodies, AW952) in blocking buffer (PBS, 0.1% Tween, 3% BSA) on rocking plate at room temperature. The sample was washed four times with PBS, 0.1% Tween before anti-mouse AlexaFluor 647 (1:200, Jackson Immunoresearch) secondary antibody and Hoechst 33342 (1:2000, Molecular Probes 10 mg/mL) was added in blocking buffer. Samples were incubated for 1 h on a rocking plate at room temperature. Then, the sample was washed four times with PBS, 0.1% Tween and imaged in PBS. All the above steps were performed gently to avoid the loss of mitotic cells.

For each condition 16 images per well were taken with on IXM confocal automatic microscope (Molecular Device) with 20x water immersion objective (0.95 NA). 8 z-stacks spanning 16 µm were recorded and the maximum projection was used for analysis. For the main figure, images were taken with a 60x objective and spinning disk confocal setup. These images were used for illustration purposes in Fig. 5b.

Image analysis was performed with MetaXpress Custom Module editor software. Briefly, cell and nuclei masks were created using Hoechst to generate the master object (cell). To identify the arrested cells, nuclei colocalizing with the BubR1 signal were counted. The fraction of arrested nuclei per image based on the total number of nuclei per image was normalized to the fraction of arrested nuclei in the DMSO condition. Statistical analysis was performed with GraphPad Prism.

## Activation in one FOV

For the activation in one Field Of View (FOV) in Figs. 5e, 6b and Supplementary Fig. 10, HeLa H2B-mCherry cells were seeded as described above in a 96-well plate and synchronized with monastrol. For uncaging, the IXM confocal automatic microscope (Molecular Device) with 20x water immersion objective (0.95 NA) was used. Activation of the inhibitor was done with the 488 nm light source focused through the objective. Activation was performed for 2, 6 and 12 seconds with 100% light power.

## Time-lapse imaging

After activation, the H2B-mCherry signal was imaged. Imaging was performed with 20x water immersion objective (0.95 NA) at 37°C with 5% CO$_2$. 8 z-stacks spanning 16 µm were recorded and the maximum projection was used for analysis. In Fig. 5e, Fig. 6b and Supplementary Fig. 10 one image was taken every 10 minutes for 120 minutes and in Supplementary Fig. 9, one image was taken every 20 minutes for 6 h.

Image analysis was performed with MetaXpress Custom Module editor software. Briefly, nuclei masks were created using H2B-mCherry signal to generate the master object (cell). Signals corresponding to dying cells were excluded by applying a top hat transformation and excluding extremely bright spots. Next, arrested cells were identified by first applying a top hat transformation and then a bottom hat transformation. A mask of those cells was created using a gaussian filter. The arrested cells were identified in the mask of all nuclei. The number of arrested cells was counted for each time point. In Figs. 5e, 6b, and Supplementary Fig. 10 the final number of arrested cells (t = 120 min) was normalized to the number of arrested cells at the

beginning of the experiment (t = 0 min). In Supplementary Fig. 9 the percentage of arrested cells normalized to the total number of cells was calculated. Out of focus movies were excluded from the analysis. Dead cells were excluded from the analysis.

### 3D Cell Culture, Imaging and Analysis

HeLa H2B-mCherry cells were grown in DMEM (ThermoFisher Scientific, 31966021) supplemented with 10% heat-inactivated FBS (Capricorn Scientific, FBS-16A) and 1x Pen-Strep (ThermoFisher Scientific, 10378016) at 37°C in humidified conditions with 5% $CO_2$. On day 0, cells were trypsinized (PAN-Biotech, P10-024100), resuspended in complete DMEM no phenol red medium (ThermoFisher Scientific, 21063029) and counted in Countess II FL automated cell counter (Invitrogen). 250 cells/well were seeded in a Nunclon™ Sphera™ 96-Well plate (ThermoFisher Scientific, 174925), centrifuged at 280 g for 5 minutes at room-temperature, and grown for 72 h. For short-term treatment, 3-day-old spheroids were incubated with DMSO (final concentration 0.2%) and **cPlk1i-COOc** (final concentration 5 μM) for 2 h prior to imaging. To improve the penetration efficiency of the doubly caged Plk1 inhibitor, 0.2% DMSO and 5 μM **cPlk1i-COOc** were added immediately after cell seeding (Day 0) and incubated for 72 h at 37°C in a 5% $CO_2$ environment.

On day 3, DMSO and **cPlk1i-COOc** were added to each well, at a final concentration of 0.5% and 5 μM, respectively. Treatment was done 2 h before imaging. The plate was kept in the dark, wrapped in aluminum foil in the incubator (37°C, 5% $CO_2$).

Live-cell imaging was performed on a temperature- and $CO_2$-controlled Zeiss LSM 980 scanning confocal microscope equipped with a Multialkali-PTM detector, and a Plan-ApoChromat 10x/0.45 NA air objective, yielding a 2.4 μm/pixel (px) sampling. The excitation optics are composed of two lasers lines 488 nm (for uncaging of **cPlk1i-COOc**) and 561 nm (for imaging of mCherry-H2B) on an inverted Microscope Axio Observer 7. Activation of the inhibitor in one quarter of the spheroid (512 × 512 px, 5x zoom) was performed with 488 nm laser, which spanned an intensity of 0.52 mW at the selected region-of-interest (ROI), for 4.1 μsec/px, throughout a stack of 10 slices (90 μm). Uncaging was performed using 8x averaging, resulting in a total uncaging time of 69 seconds per ROI. Recording of the full spheroid (1024×1024 px, 2x zoom) after uncaging of **cPlk1i-COOc** was performed using a 561 nm laser, with an intensity of 45 μW at the sample. An image stack of 10 planes was acquired every 15 minutes, spanning a total depth of 80 μm, for 12 h. To assess the compound penetration in spheroids, coumarin fluorescence was detected in 3-day spheroids using a 488 nm laser line. A central-plane image was initially taken, followed by a stack of 10 planes, spanning a total depth of 90 μm. All image acquisition parameters were controlled in Zen Blue (Zeiss).

Analysis of the uncaging experiments and penetration were carried out in Fiji (ImageJ)[67], in z-stack images. If needed, spheroid movement throughout the movie was corrected by running StackReg plugin[68] prior to quantification. Within a ROI centered on the spheroid and containing two regions (light and dark quadrants), all dividing cells were tracked, and mitotic duration determined. For that, the number of frames from mitotic entry (first frame after nuclear envelope breakdown, determined based on the increased chromosome condensation) to anaphase onset (first frame where separation of chromosomes is visible) were counted, multiplied by the time-lapse and shown in minutes. Cells in which one of these phases were not identified were excluded from the analysis. Main ROI was adjusted to follow spheroid movement/rotation. Spheroids that experienced a rotation higher than 30–40° during the movie were excluded, as adjustment of the ROI and tracking of illuminated/dark quadrants was not possible. A "null" area in the middle of the main ROI was defined within the common borders of the two analyzed quadrants, to exclude the region where uncaging could have been ambiguous. Note that quantification carries an error ± 15 minutes, corresponding to the fixed time-lapse.

Deconvolved images were obtained using Huygens and are shown in Fig. 6d (zoom). Spatial classification of all cells located within the illuminated quadrant of spheroids treated with **cPlk1i-COOc** was performed by assigning each cell to one of two spheroid regions: core region - cells located within the inner 7 z-planes, extending up to 2/3 from the center of the spheroid; and outer region - cells found within the outer 3 z-planes, and located in the outermost third of the spheroid. Fiji, Adobe Photoshop 2021 and Adobe Illustrator CS5 (Adobe Systems) were used for image histogram adjustments and panel assembly for publication. RStudio was used to analyze and plot the cumulative frequency of mitotic times.

### Software

For the NMR acquisition the software used is Bruker IconNMR 5.0.10.Build19, or TopSpin 3.6.2 and the spectra is analyzed with MestReNova v 12.0.1−20560. Microscopy images were processed and analyzed in Fiji (ImageJ)[67] or MetaExpress. Statistical analysis was performed with GraphPad Prism or RStudio. Figures were assembled in Adobe Illustrator. Chemical structures were drawn in ChemDraw 22.0.0 (PerkinElmer) as well as the schematic drawing of Plk1. The co-crystal structure (PBD 2RKU) was made in PyMOL with the data from Kothe et al. 2007[40].

### Statistics and reproducibility

For all the data of this manuscript (main and supplementary information), the number of independent biological replicas (N), the number of cells analyzed (n), and the statistical analysis performed are indicated in each figure legend. The exact p values are in the figure legends and/or in the source data file. No statistical method was used to predetermine the sample size. Out-of-focus movies were excluded from the analyses (Fig. 6e and Supplementary Fig. 9). The experiments were not randomized. The investigators were not blinded to allocation during experiments and outcome assessment.

### Reporting summary

Further information on research design is available in the Nature Portfolio Reporting Summary linked to this article.

## Data availability

The authors declare that all data supporting the findings of this study are available in the article, its supplementary files, as well as in the source data file. All raw data related to main and Supplementary Figs. are deposited in the Yareta repository https://doi.org/10.26037/yareta:os6ctilkh5d4hgqrpbdu4lk45u. Source data are provided with this paper.

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

## Acknowledgements

We are very grateful to Vincent Mercier (ACCESS platform, University of Geneva, Switzerland) for discussions, suggestions, and help in setting up experiments in the platform; Sofia Barluenga for assistance with HR-MS measurements; the NMR facility (Faculty of Science), bioimaging facilities (Faculty of Medicine and Faculty of Science) and the flow cytometry facility (Faculty of Medicine) for their services. We would like to thank Patrick Meraldi (University of Geneva) and Izabela Sumara (IGBMC – CNRS, Strasbourg, France) for reagents. BI2536 was kindly provided by Boehringer Ingelheim via its open innovation platform opnMe, available at https://opnme.com. We thank Patrick Meraldi and his group, and present and past members of the Gotta and Winssinger laboratories for discussions and suggestions. A special thanks to Luca Cirillo for introducing VG to the beauty of mitosis and to Françoise Schwager for her technical help. This work was supported by the NCCR Chemical Biology (SNF 185898 to MG and NW). Work in the MG, NW and IG laboratories is supported by the Swiss National Science Foundation (grant number 310030_204267 to MG; grant number 188406 to NW, and PCEFP3_194312 to IG) and the University of Geneva. AA is a recipient of a postdoctoral fellowship from EMBO (EMBO ALTF 258–2023 to AA).

## Author contributions

V.G. performed all experiments and contributed to the design of the work, A.A. contributed to the design, implementation, and quantification of the data in Fig. 6c–e, and in Supplementary Figs. 6, 9, and 17. D.C. contributed to the design of the synthesis route of the cPlk1i, IG contributed to the design and quantification of the data in Fig. 6c–e, V.G. and M.G. wrote the initial draft of the manuscript that was critically commented on and revised by all authors. N.W. and M.G. contributed the design, conceptualization, and methodology of the study.

## Competing interests

The authors declare no competing interests.
