## [Transparent Peer Review file · Nature Communications]

Spatio-temporal control of mitosis using light via a Plk1 inhibitor caged for activity and cellular permeability

Corresponding Author: Professor Monica Gotta

Version 1:

Reviewer comments:

Reviewer #1

(Remarks to the Author)

This manuscript by Nicolas, Monica and co-authors describes the synthesis of two caged derivatives of BI2536, a known inhibitor of the Polo-like kinase 1 (Plk1), and the study of the mitosis inhibition upon visible light irradiation, both in cell monolayers and in spheroid cultures. In both caged compounds, a 7-diethylaminocoumarin bearing an electron donating group (p-ethoxystyryl) at position 3 of the coumarin skeleton was used, which allowed cellular uncaging to be performed under the microscope by using 488 nm light. Overall, this is an interesting contribution for researchers working in photopharmacology since demonstrates the potential of using coumarin-based caging groups for controlling the activity of key cellular enzymes by taking advantage of the retention of the resulting photocage at the site of uncaging. However, although the authors state that this strategy could be “useful to control and study of cellular processes in organoids or mammalian embryos”, it is limited by the penetration capacity of the light required to trigger the uncaging process (488 nm). In this sense, the use of NIR light offers several advantages in photopharmacology, including great penetration capacity, minimal tissue absorption and scattering, and low autofluorescence. In my opinion, the manuscript should better fit in a more specialized journal.

Additional comments:

- 1.- The description of Figure 1 on page 4, particularly of panels d and e, is a little bit confusing. The purpose of including the second caging group should be better explained from the beginning, as well as the synthetic strategy (e.g. the replacement of the N-methyl group by N-Boc).
- 2.- Photolabile protecting groups (PPG) are commonly referred to as caging groups but not as photocages. Please, correct this along the manuscript. The coumarin is the PPG or caging group but not the photocage, which should be reserved for the caged compound (drug+PPG).
- 3.- The dark stability of the caged compounds should be investigated in cell culture medium by LC-MS as well as their cytotoxicity.
- 4.- Photolysis studies should be performed in PBS or in cell culture medium rather than in water. The authors indicate that uncaging leads to “an intramolecular cyclization, which changes the π -system and blue-shifts the absorption of the coumarin”. However, no evidence of the formation of this photoproduct is provided in the manuscript. In general, the chemistry should be explained in more detail in the manuscript, specially from a mechanistic point of view.
- 5.- The photophysical properties of the caged compounds as well as the uncaging quantum yield should be provided.
- 6.- Regarding cellular uptake studies by fluorescence microscopy (Figure 3c,d), I would recommend to investigate the subcellular localization of the caged compounds as well as of the coumarin photoproducts. Do they accumulate in endosomes or lysosomes in the cytoplasm?
- 7.- I would suggest the authors to study the penetration of the caged compound inside 3D spheres by fluorescence microscopy imaging.

Reviewer #2

(Remarks to the Author)

The authors of this study introduced a novel strategy to enhance the control of BI2536, a well-known inhibitor of the essential mitotic kinase Polo-like kinase 1 (Plk1). Controlling the kinase activity of Plk1 with spatial and temporal precision has intensively been pursued in the last few years and is crucial for elucidating signaling pathways governed by this kinase. By attaching a coumarin photocage to two specific positions on BI2536—one preventing interaction with Plk1 and the other

masking an added carboxylic acid for cellular retention—the inhibitor BI2536 can be activated and trapped in cells through non-invasive light exposure. This method was tested in three-dimensional spheroid cultures, demonstrating that a single light pulse could inhibit Plk1 and arrest cell division with high spatiotemporal control. The modified inhibitor proved efficient in causing mitotic arrest, showcasing its potential for precisely manipulating cellular processes. Most importantly, this novel approach can be applied to other small molecules, offering a new solution for controlling their activity in living cells with high precision.

Overall, this study has been conducted systematically, with the chemical and synthetic aspects described clearly and concisely. However, some functional and biological study shortcomings need to be addressed to strengthen the results and the authors conclusions.

Given the potential of this new formula to improve the spatiotemporal control of kinase inhibitors and the advantages it could offer to research in the field of kinase regulation, I encourage the publication of this study. However, some revisions need to be made to address the points raised. Specific comments are provided below.

Major:

- 1- Is there a specific cause for the authors use of cPLK1i at a concentration of 500 nM in Extended Figure 1, while 100 nM appears enough to induce a substantial mitotic arrest comparable to that generated by BI2536 following cPLK1i uncaging (Figure 3B)? Additionally, did the authors observe any cellular toxicity associated with this higher concentration?
- 2- Figure 4 (a, b) An increase in the mitotic index or mitotic arrest could be considered an acceptable indicator of cPLK1i reactivation following light treatment. However, it remains unproven whether PLK1 activity or its interaction with known substrates during mitosis, which is activity-dependent, such as the interaction with CENPU, is similarly affected. Given that this novel molecular tool is designed to dissect PLK1 functions in different temporal and spatial contexts during mitosis, further experiments are necessary to validate the efficacy of the proposed system. Specifically, it is essential to assess PLK1 activity using Western Blot analysis with a phospho-threonine 210-specific antibody and to examine the phosphorylation status of PLK1 targets, such as Myt1, TCTP, or Bora. Additionally, since BI2536 also influences the mitotic localization of PLK1, it is essential to determine whether the uncaging of cPLK1i induces a similar phenotype.
- 3- Figure 4 b: This figure includes boxes showing cells in interphase and mitosis. Please clarify what is to be understood from these inserts.
- 4- Figure 5: Similarly, what justifies the further increase in concentration of the double caged cPLKi COOc utilized in this experiment, from 500 nM to 1 μ M and 5 μ M? Does the presence of double caging influence the quantum yield? Moreover, how do the authors ensure the absence of off-target effects associated with these very high concentrations, particularly considering that 1 μ M BI2536 displays inhibitory activity against other members of the PLK family and also affects other mitotic kinases? Additionally, the high compound concentration, regardless of its inhibitory capacity, may induce cytotoxicity within cellular systems. This can result from accumulating the compound or its by-products, leading to cellular stress, and potentially compromising cellular viability.
- 5- Figure 5d: The spatial distribution of the illuminated and non-illuminated spheroid quadrants of the cPLKiCOOc-treated group appears quite far apart, which could affect the interpretation of the results. It is, therefore, recommended that the analyzed quadrants be positioned closer together, following the same methodology adopted in the DMSO-treated control group. This adjustment of the quadrant analysis would improve the reliability and comparability of results between experimental conditions.

Reviewer #3

(Remarks to the Author)

Studying the spatio-temporal regulation of cell division enzymes is crucial for understanding their detailed functions. The authors introduce a novel inhibitor designed to inactivate Plk1 kinase, a key regulator of cell division. This technology involves two chemical modifications to the well-known BI2536 Plk1 inhibitor, creating a modified version (cPlk1iCOOc) that remains inactive until exposed to light. Upon light irradiation, the inhibitor is "uncaged," revealing the Plk1 inhibitory site and thereby inhibiting Plk1. Additionally, light activation prevents the inhibitor from crossing the cytoplasmic membrane, ensuring that only irradiated cells are affected.

This tool offers potential for further analysis of Plk1-dependent processes and can be adapted for other inhibitors. The current functional analyses focus on mitotic progression and the quantification of mitotic indices, but more detailed phenotypic analyses are needed.

Major Points

- Comparison with BI2536: Previous studies have demonstrated that the BI2536 inhibitor effectively arrests cells during division, mimicking Plk1 loss of function. It is unclear if the light-activated cPlk1iCOOc, a derivative of BI2536, is as effective and stable over time as the original BI2536. This study does not accurately compare the two. Time-lapse experiments without pre-synchronization using the Kinesin5 inhibitor Monastrol would provide a better comparison. Is this pre-synchronization with Monastrol necessary to fully exploit the potential of this inhibitor? Generally, it would be beneficial to quantify and compare the effects of cPlk1iCOOc and BI2536 on Plk1-dependent mitotic functions.
- Cell Cycle Arrest: It is mentioned that cells treated with light-activated cPlk1iCOOc are arrested in metaphase (line 299). Are these cells truly in metaphase, or are they in prometaphase with the characteristic mitotic defects of Plk1 inhibition? This relates to the earlier point about the comparability with BI2536.
- Testing in Other Model Systems: Have the authors tested whether this Plk1 inhibitor works in other model systems?

Reviewer #4

(Remarks to the Author)

Version 2:

Reviewer comments:

Reviewer #1

(Remarks to the Author)

In this revised version, the authors have addressed most of the reviewers' comments. The manuscript improved in significance and rigor. However, there are still several critical issues that have not been addressed by the authors and that should be fixed prior to publication:

-As I noted in my previous report, "photolabile cages" or "photolabile coumarin cage" are not appropriate terms. I would suggest using "photolabile protecting groups" or "caging groups" or "coumarin photocages". The use of non-standard terminology can cause confusion for readers and non-experts in the field.

-In the rebuttal letter, the authors indicate that the coumarin PPG could be removed by two-photon irradiation to achieve high penetration. However, no attempt was made to prove this. In addition, two-photon technology, although powerful, presents several challenges compared to one-photon technology and cannot be considered a routine alternative. A comment on this issue would be appreciated by readers as well as to the challenge that supposes the development of PPGs activatable with long wavelengths.

-Regarding the uncaging quantum yield of the coumarin-based PPG used in this work, the authors indicate that it has been previously reported and provide a citation to the article. However, in my opinion, it should be calculated. The uncaging quantum yield depends on several factors, including the cargo molecule and the photolysis conditions. I encourage the authors to determine it as well as the chemical yield for the uncaging process.

Reviewer #2

(Remarks to the Author)

I thank the authors for their efforts in addressing the comments and critiques raised on their manuscript. The authors have successfully responded to all feedback, incorporating additional experiments and introducing clarifications throughout the manuscript text, which has significantly enhanced the quality of the proposed article. In its revised form, I encourage the journal to consider this article for publication.

Reviewer #3

(Remarks to the Author)

The authors have effectively addressed most of my initial concerns. As noted, the majority of the phenotypes discussed pertain to mitotic arrest, and they have provided clear explanations for these observations. However, in the comparison of phenotypes induced by cPlk1iCOOc and BI2536 on Plk1-dependent mitotic functions, I feel the paper could have been strengthened with a more thorough analysis of additional defects, particularly those related to mitotic spindle structure. This was a suggestion in my initial review, but it appears the authors did not fully explore it. Including microtubule staining and quantification of centrosomal protein recruitment would be relevant for this study, especially given Plk1's essential role in centrosome maturation. While this experiment may be considered supplementary, it is relatively straightforward and could significantly enhance the depth and impact of the study.

Minor comment: In Figure S9, the quantification of mitotic indices lacks a control for the inhibitor's effect in the absence of light (that should not trigger an increase of the mitotic index, similar to DMSO). Is the light exposure during image acquisition sufficient to activate the inhibitor, or should a non-illuminated control be included to verify the results?

Reviewer #4

(Remarks to the Author)

Version 3:

Reviewer comments:

Reviewer #1

(Remarks to the Author)

The authors have addressed all my initial concerns through rigorous revision. In my opinion, the manuscript can be considered for publication in its current form.

Reviewer #3

(Remarks to the Author)

The authors have satisfactorily answered my last questions regarding a more accurate characterization of the mitotic spindle phenotype. The manuscript in its present form can be published at Nature Communications.

Dear reviewers,

We thank you and the reviewers for the constructive criticisms on our work. As you will see in our resubmitted article, we have addressed all comments with new experiments in many cases and with text changes in some cases.

Here are four key updates to strengthen the manuscript:

- **Cell viability assessment:** We have assessed the viability of cells exposed to **cPlk1i** and **cPlk1i-COOOc**, in both caged and uncaged forms. Importantly, neither form caused a reduction in cell viability. These data are now presented in Supplementary Figures 7 and 16.
- **Plk1 specific phenotypes:** We further explored Plk1 specific phenotypes by comparing our BI2536 derived inhibitors to the original BI2536. The data are in Figures 4, 5 and Supplementary Figures 9 and 15.
- **Inhibitor localization:** We have further described the localization of the inhibitors at a subcellular level and the penetration into spheroids (Supplementary Figures 6 and 17).
- **Spatial analysis of mitotic timing in spheroids:** We also included a more extensive spatial analysis of the mitotic timing within the spheroid, comparing the regions exposed and not exposed to light (Fig. 6 and Supplementary Figure 17).

All the major changes are highlighted in yellow in the manuscript and detailed in our response letter.

REVIEWER COMMENTS

Reviewer #1 (Remarks to the Author):

This manuscript by Nicolas, Monica and co-authors describes the synthesis of two caged derivatives of BI2536, a known inhibitor of the Polo-like kinase 1 (Plk1), and the study of the mitosis inhibition upon visible light irradiation, both in cell monolayers and in spheroid cultures. In both caged compounds, a 7-diethylaminocoumarin bearing an electron donating group (p-ethoxystyryl) at position 3 of the coumarin skeleton was used, which allowed cellular uncaging to be performed under the microscope by using 488 nm light. Overall, this is an interesting contribution for researchers working in photopharmacology since demonstrates the potential of using coumarin-based caging groups for controlling the activity of key cellular enzymes by taking advantage of the retention of the resulting photocage at the site of uncaging. However, although the authors state that this strategy could be “useful to control and study of cellular processes in organoids or mammalian embryos”, it is limited by the penetration capacity of the light required to trigger the uncaging process (488 nm). In this sense, the use of NIR light offers several advantages in photopharmacology, including great penetration capacity, minimal tissue absorption and scattering, and low

autofluorescence. In my opinion, the manuscript should better fit in a more specialized journal.

This is a good point, and we agree with this reviewer that, in addition to phototoxicity, light penetration depth is another challenge in the field of photopharmacology. The coumarin that we selected for our study allows for uncaging with up to 488 nm light which penetrates up to 1 mm into tissues¹. This proved sufficient for spheroid studies. In light of this point, we now re-analyzed the data and showed that cells at different depth in the spheroid were equally inhibited. The following sentence was added to the main text to clarify this point in line 386: "Arrested cells were present in both the outer layer and the core of the spheroid (Supplementary Fig. 17c), suggesting that the compound penetrates and is activated to sufficient levels to inhibit cell division even in the spheroid's core.". It is also important to note that the coumarin that was used in this study can be uncaged with two-photon irradiation with light of 730-880 nm (i.e. NIR) as reported by Lin and coworkers². While this proved not to be necessary in the present case, the light penetration limitation can be overcome with two-photon. As reported by Lin, the *p*-ethoxystyryl derivatized coumarin has a maximum two-photon cross-section within 730-880 nm of 58.6 GM (1 GM = 10⁻⁵⁰ cm⁴s/photon) compared to 30.8 GM for DEAC and an uncaging efficiency of 26.4 GM compared to 2.16 GM for DEAC, making it one of the best in class. Finally, the choice of caging groups also needs to balance the ground state stability and size. It is not clear if the inclusion of two NIR-sensitive caging group would retain cellular permeability.

1. Finlayson, L. *et al.* Depth Penetration of Light into Skin as a Function of Wavelength from 200 to 1000 nm. *Photochem & Photobiology* **98**, 974–981 (2022).
2. Lin, Q. *et al.* Coumarin Photocaging Groups Modified with an Electron-Rich Styryl Moiety at the 3-Position: Long-Wavelength Excitation, Rapid Photolysis, and Photobleaching. *Angewandte Chemie International Edition* **57**, 3722–3726 (2018). - Reference 42 in the main text.

Additional comments:

1.- The description of Figure 1 on page 4, particularly of panels d and e, is a little bit confusing. The purpose of including the second caging group should be better explained from the beginning, as well as the synthetic strategy (e.g. the replacement of the N-methyl group by N-Boc).

We thank the reviewer for pointing out the lack of clarity. The text has been revised to mark the importance of the second caging group and we better describe Fig. 1 both in the text and in the legend. The synthetic strategy is shown in Supplementary Fig. 1 and Supplementary Fig. 13.

2.- Photolabile protecting groups (PPG) are commonly referred to as caging groups but not as photocages. Please, correct this along the manuscript. The coumarin is the PPG or caging group but not the photocage, which should be reserved for the caged compound (drug+PPG).

The reviewer is right, we have now corrected this throughout the manuscript.

3.- The dark stability of the caged compounds should be investigated in cell culture medium by LC-MS as well as their cytotoxicity.

This is a good point. We have now measured the stability of the caged compounds by LC-MS and the results are shown in Supplementary Fig. 4. The caged compounds were found to be stable for 24 hours in supplemented cell culture medium.

We had not observed cell death nor slow growth in any of our experiments. Based on the request of this and of the reviewer 2, we have quantified cytotoxicity with both the caged and the uncaged compounds with a live-dead staining assay. We assessed the toxicity of the uncaged compounds after 18 hours of incubation, which is 6 hours longer than the incubation we used in our studies (see spheroid experiment). The commercial Plk1 inhibitor BI2536, on which our compound is based, shows some toxicity at 1 pM after 16 hours of incubation. Neither **cPlk1i** (1 pM) nor **cPlk1i-COOC** (5 pM) showed cell death 16 hours after the activating light pulse, even though the cells are arrested in mitosis. This is comparable to 100 nM BI2536 in the same panel. These data are now shown in Supplementary Fig. 7 and Supplementary Fig. 16 and lines 207 and 350.

For the caged compound we extended the incubation time to 24 hours. Neither **cPlk1i** up to 1 pM nor **cPlk1i-COOC** up to 5 pM showed decreased cell viability (Supplementary Fig. 7 and Supplementary Fig. 16).

4.- Photolysis studies should be performed in PBS or in cell culture medium rather than in water. The authors indicate that uncaging leads to "an intramolecular cyclization, which changes the n-system and blue-shifts the absorption of the coumarin". However, no evidence of the formation of this photoproduct is provided in the manuscript. In general, the chemistry should be explained in more detail in the manuscript, specially from a mechanistic point of view.

As suggested by this reviewer we have now performed the photolysis study in PBS. The results are shown in Supplementary Fig. 3 and 14b and we found comparable performance.

The intramolecular cyclization and blue-shift in absorption has been reported by Lin et al. The NMR data in our study corroborates this finding with the disappearance of olefin (d^*) after irradiation with light (see Figure below). However, we are not claiming this is new and to make it abundantly clear that this was first reported by Lin, another reference was added after the sentence "Since uncaging leads to intramolecular cyclization, .." line 234.

Figure: NMR spectra over time of illumination with 455 nm light.

5.- The photophysical properties of the caged compounds as well as the uncaging quantum yield should be provided.

The uncaging quantum yield of the coumarin used in this study has been reported previously by Lin et al. and cited in the manuscript. To clarify our motivation with values we have now added the quantum yield and maximal absorption wavelength for the coumarin used in this study and the reference coumarin (DEAC). The text now reads in line 133: "As photolabile protecting group, we selected the 7-diethylaminocoumarin with the electron donating p-ethoxystyryl moiety in the 3-position (Fig. 2)⁴². This cage has an absorption that extends to 500 nm (λ_{\max} at 439 nm) with a high uncaging quantum yield Φ_u of 0.45 compared to 0.07 for the commonly used 7-diethylaminocoumarin (DEAC) cage ($\lambda_{\max} < 400$ nm)."

6.- Regarding cellular uptake studies by fluorescence microscopy (Figure 3c,d), I would recommend to investigate the subcellular localization of the caged compounds as well as of the coumarin photoproducts. Do they accumulate in endosomes or lysosomes in the cytoplasm?

To assess whether **cPIk1i** and **cPIk1i-COOC** accumulate in endosomes or lysosomes, we imaged cells treated with **cPIk1i** or **cPIk1i-COOC** and incubated with LysoTracker which stains for late endosomes and lysosomes. We did not observe colocalization of the inhibitors with the LysoTracker stained organelles indicating that the inhibitors are not specifically accumulating in late endosomes or lysosomes. We included this data in

Supplementary Fig. 6. It should be noted that for the doubly caged compound (**cPlk1i-COOc**), since it is active once uncaged and does not diffuse through membranes, a significant fraction of the caged compound must be in the cytosol (since if it was restricted to vesicles, it would not be able to inhibit Plk1). The localization of the coumarin byproduct was not investigated and presumably diffuses out of the cell considering its low molecular weight and the absence of charges. Its relatively weak fluorescence at 400 nm renders analysis difficult considering the high background fluorescence at the excitation wavelength.

7.- I would suggest the authors to study the penetration of the caged compound inside 3D spheres by fluorescence microscopy imaging.

We have addressed this question in two ways.

In order to investigate the penetration efficiency of the doubly-caged compound, we treated spheroids for 2 and 72 hours and followed the coumarin fluorescence by live microscopy. In the former case, although the middle z-plane displayed fluorescence at the center of the spheroid, the signal was not homogeneous, being higher at the outer region of the spheroid (Supplementary Fig. 17a). This was significantly improved by incubating the spheroids with **cPlk1i-COOc** (5 μ M) for 72 hours (added at day 0). On the other hand, treatment with the DNA dye SPY505-DNA for 2 hours led to a fluorescence gradient that varies along the radius of the sphere similar to what we observed with 2 hours treatment with **cPlk1i-COOc** (5 μ M) (Supplementary Fig. 17b). This result suggests that the partial compound penetration observed after short incubation periods is not specific to **cPlk1i-COOc**, but rather a limitation of large 3D-cell aggregates.

To further validate that **cPlk1i-COOc** diffused throughout the spheroid, we assessed the presence of arrested cells in mitosis in the different spheroid regions (outer and inner layers). Delayed and arrested cells were identified regardless of their position in the spheroid (Supplementary Fig. 17c), suggesting that the inhibitor penetrates to the core of the spheroids, and it can be activated at this location.

We have now included the following sentence in the main text in line 386: "Arrested cells were present in both the outer layer and the core of the spheroid (Supplementary Fig. 17c), suggesting that the compound penetrates and is activated to sufficient levels to inhibit cell division even in the spheroid's core."

Reviewer #2 (Remarks to the Author):

The authors of this study introduced a novel strategy to enhance the control of BI2536, a well-known inhibitor of the essential mitotic kinase Polo-like kinase 1 (Plk1). Controlling the kinase activity of Plk1 with spatial and temporal precision has intensively been pursued in the last few years and is crucial for elucidating signaling pathways governed by this kinase. By attaching a coumarin photocage to two specific positions on BI2536—one preventing interaction with Plk1 and the other masking an added carboxylic acid for cellular retention—the inhibitor BI2536 can be activated and trapped in cells through non-invasive light exposure. This method was tested in three-

dimensional spheroid cultures, demonstrating that a single light pulse could inhibit Plk1 and arrest cell division with high spatiotemporal control. The modified inhibitor proved efficient in causing mitotic arrest, showcasing its potential for precisely manipulating cellular processes. Most importantly, this novel approach can be applied to other small molecules, offering a new solution for controlling their activity in living cells with high precision.

Overall, this study has been conducted systematically, with the chemical and synthetic aspects described clearly and concisely. However, some functional and biological study shortcomings need to be addressed to strengthen the results and the authors conclusions.

Given the potential of this new formula to improve the spatiotemporal control of kinase inhibitors and the advantages it could offer to research in the field of kinase regulation, I encourage the publication of this study. However, some revisions need to be made to address the points raised. Specific comments are provided below.

Major:

1- Is there a specific cause for the authors use of cPLK1i at a concentration of 500 nM in Extended Figure 1, while 100 nM appears enough to induce a substantial mitotic arrest comparable to that generated by BI2536 following cPLK1i uncaging (Figure 3B)? Additionally, did the authors observe any cellular toxicity associated with this higher concentration?

We thank the reviewer for this comment that has helped us to clarify the text. The concentration needed to inhibit Plk1 can be achieved in different ways: by treating the cells with 100 nM **cPlk1i** and uncaging 100% of the compound or by increasing the concentration of the caged compound and only uncaging a fraction of it. We opted for the second strategy, especially as we had not observed cell division arrest (see cell cycle analysis in Fig. 4 a and b in the paper) or cell death (now quantified in Supplementary Fig. 7 and 16) with our caged compound up to 1 μ M. In the above-mentioned case (500 nM) uncaging 20% of the compound should result in the desired activity. As shown in Fig. 4 a and b, where the compound was uncaged before applying it to the cells, the active inhibitor is arresting cells in mitosis at 100 nM. When uncaging under the microscope we decided to increase the concentration so that there would be enough active inhibitor present in the medium after the uncaging pulse. As seen in Supplementary Fig. 8, the fluorescence of the caged coumarin decreases by roughly 40% after uncaging under the microscope (488 nm, 5.5 mW/cm²), which indicates that the concentration of active inhibitor is roughly 200 nM as observed by the increase in mitotic cells compared to cells not illuminated.

As also requested by reviewer 1, we have now performed cellular toxicity assays with both the caged and the activated compounds and did not observe cellular toxicity at any of the concentrations used in this study (see Supplementary Fig. 7 and 16 and response to Reviewer 1 - Point 3).

To clarify this point in the text, we have added the following sentence in the manuscript in line 229: "As treatment with 1 μ M **cPlk1i** did not result in toxicity (Supplementary Fig. 7b), we decided to increase the concentration of **cPlk1i** compared to BI2536. This

allowed us to use a shorter uncaging light pulse, since only a portion of the inhibitor needs to be uncaged to reach the biologically relevant concentration.”

2- Figure 4 (a, b) An increase in the mitotic index or mitotic arrest could be considered an acceptable indicator of cPLK1i reactivation following light treatment. However, it remains unproven whether PLK1 activity or its interaction with known substrates during mitosis, which is activity-dependent, such as the interaction with CENPU, is similarly affected. Given that this novel molecular tool is designed to dissect PLK1 functions in different temporal and spatial contexts during mitosis, further experiments are necessary to validate the efficacy of the proposed system. Specifically, it is essential to assess PLK1 activity using Western Blot analysis with a phospho-threonine 210-specific antibody and to examine the phosphorylation status of PLK1 targets, such as Myt1, TCTP, or Bora. Additionally, since BI2536 also influences the mitotic localization of PLK1, it is essential to determine whether the uncaging of cPLK1i induces a similar phenotype.

To address this point, we first performed western blot analysis. Despite using phospho-threonine 210-specific antibodies acquired from different companies, we could not observe a clear result, neither with treatment with the original BI2536 itself nor with our compounds. Because of this, we decided to address this question by immunostaining. Since BI2536 is an ATP competitive inhibitor, treatment with our inhibitor, which is a modification of BI2536, should result in decreased phosphorylation of Plk1 targets. BubR1 is a kinetochore protein, which is a known direct target of Plk1 and becomes phosphorylated by Plk1 on Serine 676 (<https://doi/10.1101/gad.436007>). Inhibition with BI2536 as well as treatment with **cPlk1i-COOc** and activation with light led to a decrease in S676 phosphorylation (Supplementary Fig. 15c, d). Treatment with **cPlk1i-COOc** also resulted in displacement of Plk1 from kinetochore (Supplementary Figure 15a, b), consistent with results published for BI2536 (<https://doi/10.1016/j.cub.2006.12.046>). These results show that **cPlk1i-COOc** causes the same phenotypes induced by treatment with BI2536, consistent with the fact that it derives from this compound.

1. Elowe, S., Hümmel, S., Uldschmid, A., Li, X. & Nigg, E. A. Tension-sensitive Plk1 phosphorylation on BubR1 regulates the stability of kinetochore microtubule interactions. *Genes & development* **21**, 2205–2219 (2007).
2. Lénárt, P. *et al.* The Small-Molecule Inhibitor BI 2536 Reveals Novel Insights into Mitotic Roles of Polo-like Kinase 1. *Current Biology* **17**, 304–315 (2007).

3- Figure 4 b: This figure includes boxes showing cells in interphase and mitosis. Please clarify what is to be understood from these inserts.

The reviewer is correct, we had not mentioned why some cells were in boxes in the figure legend. Initially the boxes were simply meant to show an example of a mitotic cell stained with BubR1 and the absence of BubR1 in interphase cells. For clarity, we decided to remove the boxes in now Fig. 5b and instead we highlighted the BubR1 positive cells with an arrow in the top panel of the figure.

4- Figure 5: Similarly, what justifies the further increase in concentration of the double caged cPLKi COOc utilized in this experiment, from 500 nM to 1 pM and 5 pM? Does the presence of double caging influence the quantum yield? Moreover, how do the authors ensure the absence of off-target effects associated with these very high concentrations, particularly considering that 1 pM BI2536 displays inhibitory activity against other members of the PLK family and also affects other mitotic kinases? Additionally, the high compound concentration, regardless of its inhibitory capacity, may induce cytotoxicity within cellular systems. This can result from accumulating the compound or its by-products, leading to cellular stress, and potentially compromising cellular viability.

As explained above (Reviewer 2, Point 1), we increased the concentration to be able to decrease the illumination time. When looking at the time series (Fig. 6b and Supplementary Fig. 10) we observed that uncaging for 12 seconds is needed to induce mitotic arrest with **cPlk1i-COOc** (1 pM). When increasing the concentration to 5 pM, 2 seconds of uncaging are sufficient to arrest the cells. This underlines that the concentration of the active inhibitor in the cell is directly proportional to the illumination time. For the experiments in spheroids, we increased the concentration to ensure deep penetration of the compound inside the spheroid.

As explained above (Reviewer 1, Point 3) and shown in Supplementary Fig. 7 and 16, **cPlk1i** and **cPlk1i-COOc** before and after activation with light do not lead to cell death at the concentrations that we used in this study.

Concerning the off targeted effects, BI2536 specifically inhibits Plk1 ($IC_{50}=0.83$ nM). It does however significantly inhibit the closely related members of the Plk family Plk2 ($IC_{50}=3.5$ nM, 4-fold higher than Plk1) and Plk3 ($IC_{50}=9$ nM, 11-fold higher than Plk1), as previously shown (<https://doi.org/10.1016/j.cub.2006.12.046>)³. In different studies BI2536 has been used to study the functions of these proteins and the caged inhibitors presented here could also be used to study Plk2 (<https://doi.org/10.1371/journal.pone.0009849>)⁴ or Plk3 (<https://doi.org/10.1038/cr.2016.78>)⁵. Plk2 and Plk3 have specific functions in interphase. In our study we focused to show the applicability of this novel tool compound on a phenotype related to Plk1, cell division arrest. When studying non-synchronized cells, we also observed mitotic phenotypes, suggesting that Plk1 is the main kinase that is inhibited in our experiments. To clarify we have now added the following sentence in the introduction in line 59: "BI2536 specifically inhibits Plk1 with a 1000-fold selectivity over other kinases. It also inhibits other members of the Polo-like kinase family, Plk2 and Plk3, which are mostly active in interphase"²⁴. As a tool compound, BI2536 has been fundamental in the characterization of many functions of Plk1^{28,29} as well as Plk2³⁰ and Plk3³¹"

3. Lénárt, P. *et al.* The Small-Molecule Inhibitor BI 2536 Reveals Novel Insights into Mitotic Roles of Polo-like Kinase 1. *Current Biology* **17**, 304–315 (2007).
4. Krause, A. & Hoffmann, I. Polo-Like Kinase 2-Dependent Phosphorylation of NPM/B23 on Serine 4 Triggers Centriole Duplication. *PLoS ONE* **5**, e9849 (2010).
5. Helmke, C. *et al.* Ligand stimulation of CD95 induces activation of Plk3 followed by phosphorylation of caspase-8. *Cell Res* **26**, 914–934 (2016).

5- Figure 5d: The spatial distribution of the illuminated and non-illuminated spheroid quadrants of the cPlk1COOc-treated group appears quite far apart, which could affect the interpretation of the results. It is, therefore, recommended that the analyzed quadrants be positioned closer together, following the same methodology adopted in the DMSO-treated control group. This adjustment of the quadrant analysis would improve the reliability and comparability of results between experimental conditions.

Following the suggestion of the reviewer, we analyzed the dark quadrant adjacent to the illuminated quadrant (+light). The results show that the mitotic timing of cells from the two dark quadrants (II and III) treated with cPlk1i-COOc or DMSO do not change (Figure below in this response letter). For simplicity and to improve the comparability of the results, the new dark quadrant (II) has now replaced the previous dark one (III) in the now main Fig. 6e.

To further clarify the quadrants selected for the quantification, we now included dotted boxes that delineate those areas in the images in Fig. 6d. The individual highlighted cells are representative examples of the phenotypes observed upon the different treatments, and it is now explained in the legend.

Reviewer #3 (Remarks to the Author):

Studying the spatio-temporal regulation of cell division enzymes is crucial for understanding their detailed functions. The authors introduce a novel inhibitor designed to inactivate Plk1 kinase, a key regulator of cell division. This technology involves two chemical modifications to the well-known BI2536 Plk1 inhibitor, creating a modified version (cPlk1iCOOc) that remains inactive until exposed to light. Upon light irradiation, the inhibitor is "uncaged," revealing the Plk1 inhibitory site and thereby inhibiting Plk1. Additionally, light activation prevents the inhibitor from crossing the cytoplasmic membrane, ensuring that only irradiated cells are affected.

This tool offers potential for further analysis of Plk1-dependent processes and can be adapted for other inhibitors. The current functional analyses focus on mitotic progression and the quantification of mitotic indices, but more detailed phenotypic

analyses are needed.

Major Points

- Comparison with BI2536: Previous studies have demonstrated that the BI2536 inhibitor effectively arrests cells during division, mimicking Plk1 loss of function. It is unclear if the light-activated cPlk1iCOOc, a derivative of BI2536, is as effective and stable over time as the original BI2536. This study does not accurately compare the two. Time-lapse experiments without pre-synchronization using the Kinesin5 inhibitor Monastrol would provide a better comparison. Is this pre-synchronization with Monastrol necessary to fully exploit the potential of this inhibitor? Generally, it would be beneficial to quantify and compare the effects of cPlk1iCOOc and BI2536 on Plk1-dependent mitotic functions.

To address the point on the stability, as also suggested by reviewer 1, we have included a stability assay in Supplementary Fig. 4 for both **cPlk1i** and **cPlk1i-COOc**. As shown in the figure, the compounds are stable over time (24 hours).

Concerning the comparison with BI2536, we had used our compounds and in parallel BI2536 in several assays:

- In the flow cytometry assay, we assessed both the active Boc-derivatized inhibitor (**Plk1i**) and the activated **cPlk1i** and compared with BI2536. We did not observe a difference in cell cycle arrest (Fig. 4a, b in the main text). In this experiment, cells have not been synchronized.
- When uncaging **cPlk1i** under the microscope in Fig. 5, we see cell cycle arrest in mitosis for the active inhibitor. As mentioned above (Reviewer 2, Point 1), and also now included in the main text in line 229, the increase in concentration (compared to BI2536) is because we do not uncage 100% of the inhibitor.

To further address this point, we now included two additional experiments where we compared our compounds and BI2536, as suggested by the reviewer.

- We performed time-lapse microscopy counting the number of cells in mitosis at each time point after activation of the caged inhibitors comparing it to direct addition of BI2536 over the course of 6 hours in a non-synchronized population of cells. In BI2536, as well as in uncaged **cPlk1i** and **cPlk1i-COOc** the number of mitotic cells gradually increases over the course of the experiment, which stems from a mitotic arrest due to Plk1 inhibition. In DMSO treated and illuminated conditions the number of mitotic cells is constantly low and does not increase over time as the cells proceed through the cell cycle. These results are shown in Supplementary Fig. 9. This is also visible in the spheroid experiment where the cells have not been synchronized previously (Fig. 6d, e).
- As mentioned in response to reviewer 2 (Point 2), we assessed phosphorylation of the Plk1 target p676-BubR1 and the localization of Plk1 in cells treated with **cPlk1i-COOc** and BI2536 and observed similar results (reduced phosphorylation of BubR1 and kinetochore displacement of Plk1) (Supplementary Fig. 15).

- Cell Cycle Arrest: It is mentioned that cells treated with light-activated cPlk1iCOOc are arrested in metaphase (line 299). Are these cells truly in metaphase, or are they in prometaphase with the characteristic mitotic defects of Plk1 inhibition? This relates to the earlier point about the comparability with BI2536.

We thank the reviewer for highlighting this mistake. We observe prometaphase arrested cells, as it is seen in the microscopy images in Fig. 5 and 6 and Supplementary Fig. 15. We call these arrested cells throughout the manuscript. We have now corrected this mistake in the manuscript.

- Testing in Other Model Systems: Have the authors tested whether this Plk1 inhibitor works in other model systems?

We agree with the reviewers that this is an exciting opportunity, but applying this methodology to another model systems requires a significant time investment that is beyond the scope of the present manuscript. We believe we have rigorously demonstrated that we cage both the inhibitory function and the diffusion of an important tool compound.

Reviewer #4 (Remarks to the Author):

We thank the reviewers for all the valuable comments.

Response Letter

Reviewer #1 (Remarks to the Author):

In this revised version, the authors have addresses most of the reviewers' comments. The manuscript improved in significance and rigor. However, there are still several critical issues that have not been addressed by the authors and that should be fixed prior to publication:

-As I noted in my previous report, "photolabile cages" or "photolabile coumarin cage" are not appropriate terms. I would suggest using "photolabile protecting groups" or "caging groups" or "coumarin photocages". The use of non-standard terminology can cause confusion for readers and non-experts in the field.

The manuscript has been thoroughly revised according to the reviewer's recommendation. We agree that different terminologies can be found in the literature which can be confusing. We now use both 'PPG' and 'coumarin photocage', the most used terminology. We are thus confident that it will be understood by readers from different fields (PPG being mostly used by chemists while the caging terminology having its roots in biology).

-In the rebuttal letter, the authors indicate that the coumarin PPG could be removed by two-photon irradiation to achieve high penetration. However, no attempt was made to prove this. In addition, two-photon technology, although powerful, presents several challenges compared to one-photon technology and cannot be considered a routine alternative. A comment on this issue would be appreciated by readers as well as to the challenge that supposes the development of PPGs activatable with long wavelenghts.

To address this comment, we have now added the following sentence to the discussion, line 429:

One limitation of photolabile protecting groups is the penetration in tissues of the light required to uncage. While light penetration was not a limiting factor in the present study, it has been shown that the coumarin that we used can be cleaved with two-photon irradiation with light of 730-880 nm⁴², which can overcome light penetration limitations. However, two-photon activation also presents challenges such as small irradiation area and significant heat generation. Alternatively, PPGs activatable with longer wavelengths would be beneficial for further biological applications.

-Regarding the uncaging quantum yield of the coumarin-based PPG used in this work, the authors indicate that it has been previously reported and provide a citation to the article. However, in my opinion, it should be calculated. The uncaging quantum yield depends on several factors, including the cargo molecule and the photolysis conditions. I encourage the authors to determine it as well as the chemical yield for the uncaging process.

We agree with the reviewer that the quantum yield depends on several factors, with the local cellular environment likely playing a significant role. In that respect, a quantum yield measured under ideal conditions is not always predictive of the performance in complex biological setting. The reality is a complex

environment which is not entirely aqueous and a compound that can partition in membrane or at the surface of proteins with unique electrostatic environment. Furthermore, a portion of the photons in complex biological environment are captured by other chromophores than the PPG, such as flavins, which are present in cells. We appreciate the reviewer's interest in getting more insights into the yield of the photolysis under the experimental conditions used and agree that it is more informative for a reader to understand what irradiation time and intensity is necessary to achieve a sought phenotype, i.e. yield of uncaging. Given the change of fluorescence of the coumarin after photolysis, we now used the fluorescence at $t = 0$ and the fluorescence of reaction at $t = 600$ sec (full conversion, verified by LC-MS) to calculate a half-life of reaction, hence a reaction rate. Using a 5.5 mW/cm^2 irradiation at 488 nm, $t_{1/2}=64$ seconds (Supplementary Figure 8b, c), corroborating the results from supplementary Fig. 8a. We would like to also point out that we demonstrate that under irradiation conditions that are routinely used for imaging in cellular biology, we achieve a sufficient yield to observe the phenotype resulting from inhibition of Plk1. We also show that it is possible to increase the concentration of the compound to reduce the illumination time (Supplementary Fig. 10). Since the fluorescence intensity of cPlk1i-COOc is much lower (see Supplementary Fig. 6), we were not able to perform a similar experiment with the doubly caged compound. However, from the uncaging experiments done in PBS (Supplementary Fig. 3 and Supplementary Fig. 14b), we know that cPlk1i-COOc has similar uncaging kinetics to cPlk1i. This strongly suggests that a similar time of uncaging is needed for cPlk1i-COOc.

Reviewer #2 (Remarks to the Author):

I thank the authors for their efforts in addressing the comments and critiques raised on their manuscript. The authors have successfully responded to all feedback, incorporating additional experiments and introducing clarifications throughout the manuscript text, which has significantly enhanced the quality of the proposed article. In its revised form, I encourage the journal to consider this article for publication.

We thank the reviewer for their positive feedback.

Reviewer #3 (Remarks to the Author):

The authors have effectively addressed most of my initial concerns. As noted, the majority of the phenotypes discussed pertain to mitotic arrest, and they have provided clear explanations for these observations. However, in the comparison of phenotypes induced by cPlk1iCOOc and BI2536 on Plk1-dependent mitotic functions, I feel the paper could have been strengthened with a more thorough analysis of additional defects, particularly those related to mitotic spindle structure. This was a suggestion in my initial review, but it appears the authors did not fully explore it. Including microtubule staining and quantification of centrosomal protein recruitment would be relevant for this study, especially given Plk1's essential role in centrosome maturation. While this experiment may be considered supplementary, it is relatively straightforward and could significantly enhance the depth and impact of the study.

In the first review, the reviewer asked “Generally, it would be beneficial to quantify and compare the effects of cPlk1iCOOc and BI2536 on Plk1-dependent mitotic functions.”. This is why to compare the effect of our BI2536 derivative with BI2536 we used as a marker BubR1 phosphorylated on S676. This site is specifically phosphorylated by Plk1 itself as shown by Elowe et al. 2007 and is not phosphorylated when Plk1 is inhibited by BI2536 (Elowe et al., 2007). We also showed that Plk1 is removed from kinetochores when cells are treated with our compound and with BI2536 (as previously shown with BI2536 (Lénárt et al., 2007)).

To further address this point we have now imaged mitotic spindles, which are monopolar when Plk1 is inhibited (Lénárt et al., 2007) and γ -tubulin, which is lower at centrosomes when Plk1 is inhibited (Lénárt et al., 2007). As shown in Supplementary Fig. 15a-c, and discussed in line 333-336, treatment with cPlk1i-COOc results in increased number of monopolar spindles and in reduced levels of centrosomal γ -tubulin.

Elowe, S., Hümmer, S., Uldschmid, A., Li, X. & Nigg, E. A. Tension-sensitive Plk1 phosphorylation on BubR1 regulates the stability of kinetochore microtubule interactions. *Genes Dev.* **21**, 2205–2219 (2007).

Lénárt, P. *et al.* The Small-Molecule Inhibitor BI 2536 Reveals Novel Insights into Mitotic Roles of Polo-like Kinase 1. *Curr. Biol.* **17**, 304–315 (2007).

Minor comment: In Figure S9, the quantification of mitotic indices lacks a control for the inhibitor’s effect in the absence of light (that should not trigger an increase of the mitotic index, similar to DMSO). Is the light exposure during image acquisition sufficient to activate the inhibitor, or should a non-illuminated control be included to verify the results?

We thank the reviewer for this question and take the opportunity to clarify the experimental setup. The experiment in Supplementary Fig. 9 was performed in the same way as the experiments in Fig. 5d,e and 6b and Supplementary Fig. 10. These experiments show that the imaging light (in the red channel, 560/32 nm) does not activate the inhibitor, not even in the spheroid experiment (Fig. 6c-e) where we imaged with 587 nm light over the course of 12 hours in the presence of the compound.

In Supplementary Fig. 9 we have performed time-lapse experiments without synchronization, as requested by this reviewer, and we were technically limited in the number of conditions we were able to image in parallel to have a sufficient time resolution to quantify mitosis.

Reviewer #4 (Remarks to the Author):

We thank the reviewer for the feedback.